

# The sensitivity of the Eocene-Oligocene Southern Ocean to strength and position of wind stress

Qianjiang Xing[1, 2], Dave Munday[3], Andreas Klocker[2, 4, 5], Isabel Sauermilch[2, 6], Joanne M. Whittaker[2]

[1]CSIRO-UTAS Quantitative Marine Sciences PhD Program, Institute for Marine and Antarctic Studies, University of Tasmania, Hobart, Tasmania, Australia
[2]Institute for Marine and Antarctic Studies, University of Tasmania, Hobart, Australia.
[3]British Antarctic Survey, Cambridge, United Kingdom.
[4]Australian Research Council Centre of Excellence for Climate Extremes, University of Tasmania, Hobart, Australia.
[5]Department of Geosciences, University of Oslo, Oslo, Norway
[6]Department of Earth Sciences, Utrecht University, Utrecht, The Netherlands.

*Correspondence to:* Qianjiang Xing (qianjiang.xing@utas.edu.au)

## Abstract

The early Cenozoic opening of the Tasmanian Gateway (TG) and Drake Passage (DP), alongside the synergistic action of the westerly winds, led to a Southern Ocean transition from large, subpolar gyres to the onset of the Antarctic Circumpolar Current (ACC). However, the impact of changing latitudinal position and strength of the wind stress in altering the early Southern Ocean circulation have been poorly addressed. Here, we use an eddy-permitting ocean model (0.25°) with realistic Late Eocene paleo-bathymetry to investigate the sensitivity of the Southern Ocean to paleo-latitudinal migrations (relative to the gateways) and strengthening of the wind stress. We find that southward wind stress shifts of 5 or 10°, with a shallow TG (300 m), lead to dominance of subtropical waters in the high latitudes and further warming of the Antarctic coast (increase by 2°C). Southward migrations of wind stress with a deep TG (1500 m) cause the shrinking of the subpolar gyres and cooling of the surface waters in the Southern Ocean (decrease by 3-4°C). With a 1500 m deep TG, and maximum westerly winds aligning with both the TG and DP, we observe a proto-ACC with a transport of ~47.9 Sv. This impedes the meridional transport of warm subtropical waters to Antarctic coast, thus laying a foundation for thermal isolation of the Antarctic. Intriguingly, proto-ACC flow through the TG is much more sensitive to strengthened wind stress compared to the DP. We suggest that topographic form stress can balance surface wind stress at depth to support the proto-ACC while the sensitivity of the transport is likely associated with the momentum budget between wind stress and near-surface topographic form stress driven by the subtropical gyres. In summary, this study proposes that the thermal isolation of Antarctica is a consequence of a combination of gateway deepening and the alignment of maximum wind stress with both gateways.

Key words: Southern Ocean gateways, proto-ACC, momentum balance, topographic form stress, wind-driven gyres



## 1. Introduction

The Southern Ocean is the only ocean basin without continental barriers blocking zonal
connections at all latitudes. This allows for the connection of the Pacific, Atlantic, and Indian
ocean basins by the circulation of the Antarctic Circumpolar Current (ACC), the strongest
ocean current, with a transport in the range of $137\pm7$ Sv (1 Sv=$10^6$ m$^3$s$^{-1}$) (Meredith et al.,
2011). Today, and in the geological past, two main Southern Ocean gateways are crucial for
unblocked circumpolar flow: the Tasmanian Gateway (TG, between Tasmania, Australia and
Cape Adare, Antarctica) and Drake Passage (DP, between Cape Horn and the Antarctic
Peninsula), see Figure 1. The opening (widening and/or deepening) of these two key gateways
has long been hypothesized to initiate the onset of the ACC (Kennett, 1977).

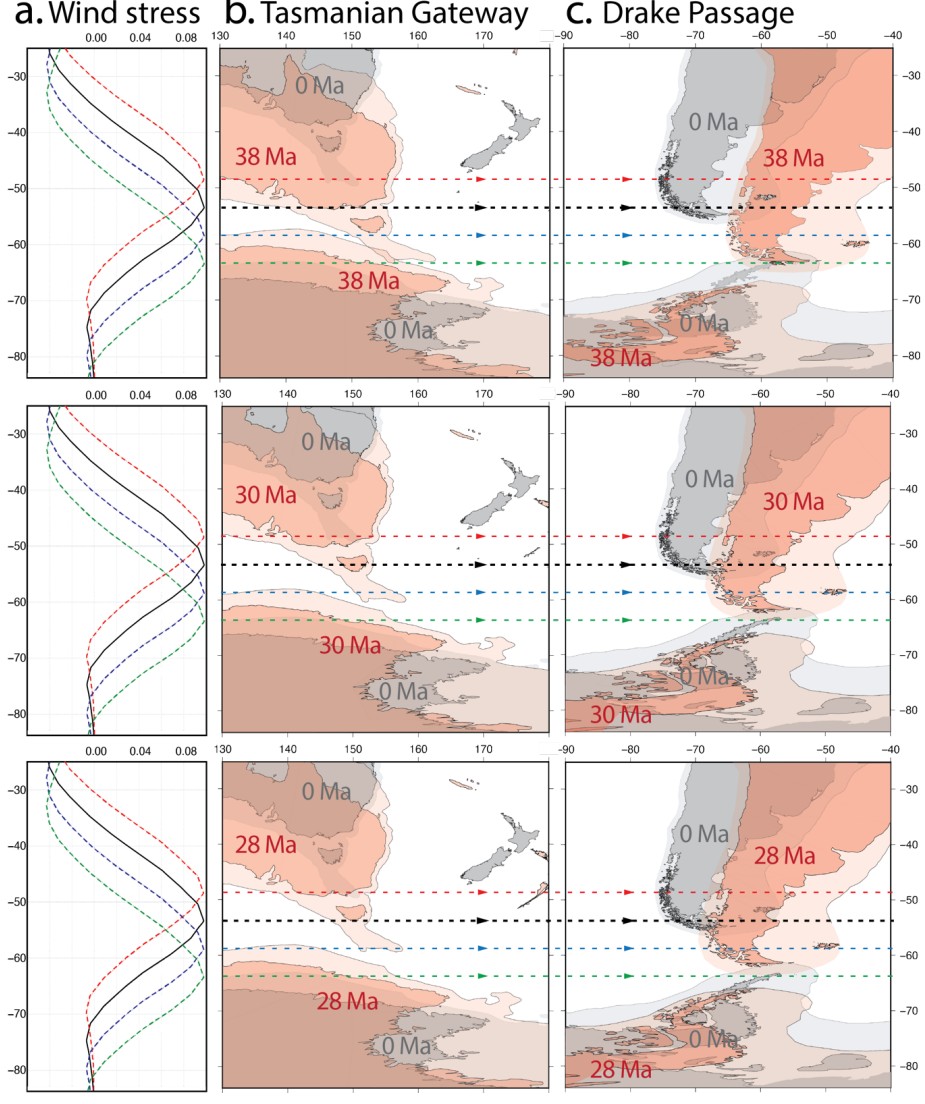



Figure 1. (a) Zonal mean wind stresses used in all simulations. Black curve (max 53°S) is the revised version from Sauermilch et al. (2021). The maximum westerly of max 53°S has strength of about 0.1 N m$^{-2}$ and locates at about 53°S. Red (max 48°S), blue (max 58°S), and green (max 63°S) curves indicate the shifts of the max 53°S curve 5 degrees to the north, 5 degrees to the south, and 10 degrees to the south, respectively. Relative positions of continents (at 38 Ma, 30 Ma, 28 Ma, and present-day) versus peak wind stresses for (b) Tasmania Gateway and (c) Drake Passage. Colored dashed lines in the (b) and (c) show latitudinal positions of the maximum westerly in four wind stress conditions. Deep orange and deep grey areas show paleo and modern continental coastline, respectively. Light orange and light grey areas represent paleo and modern continent-ocean-boundary, respectively. These reconstructions use the rotation model by Matthews et al. (2016b) with the paleomagnetic reference frame (Van Hinsbergen et al., 2015b).

## 1.1 The opening of the Southern Ocean gateways and developing ACC

The initial formation of the TG was the outcome of continental motion between Australia and Antarctica. This slow separation, over millions of years, enabled a gradual widening and deepening of the TG. A shallow TG first allowed a seawater connection from the Pacific to the Indian Ocean from ~50-49 Ma (Bijl et al., 2013). As Australia moved farther north, oceanic crust formed south of Tasmania (Royer and Rollet, 1997), with the South Tasman Rise finally clearing the Antarctic continent by 35-32 Ma. A rapid subsidence, forming a deep TG at ~35-32 Ma has been proposed based on interpretations of sediments to the east and south of Tasmania (Stickley et al., 2004). However, more recent work has suggested that the subsidence history of the TG remains unclear (e.g. Scher et al. 2015), with a lack of evidence for rapid subsidence of the East Tasman Plateau. In contrast, the evolution of the DP remains widely debated. Analyses of magnetic anomalies in the Scotia Sea, and adjacent regions, showed an initial DP opening from as early as the early Eocene, about 50 Ma (e.g. Scher and Martin (2006); Livermore et al. (2007); Van De Lagemaat et al. (2021)). A gradual deepening of the DP was inferred from the early Oligocene (Kennett, 1977) to the late Oligocene (~26 Ma) for an intermediate water exchange (Barker and Thomas, 2004).

The role of the opening and deepening/widening of the Southern Ocean gateways in altering ocean circulation and Antarctic climate is widely discussed (Kennett, 1977; Huber et al., 2004; Stickley et al., 2004; Lyle et al., 2007; Bijl et al., 2013; Sauermilch et al., 2021). A key hypothesis is that the opening gateways caused cooling of the Southern Ocean and Antarctica due to the ACC onset, by thermally isolating Antarctica from warm subtropical waters. However, oceanographic models have struggled to reconstruct the required Eocene high latitude ocean warmth prior to ACC establishment. For example, Huber et al. (2004) reconstruct the Eocene circulation of the Southern Ocean and find that the East Australian Current (EAC) flowed poleward in the mid-latitude Pacific. It is then deflected eastward from the north-eastern margin of Australia, rather than reaching high latitudes. Thus, Huber et al. (2004) proposed that insufficient warm water from the subtropics reached high latitudes to keep a pre-Eocene Antarctica warm.

The role of gateway opening on the ocean heat transport to the Antarctic coast has recently been revisited by Sauermilch et al. (2021) using eddy-permitting model simulations with realistic paleo-bathymetry. With only one gateway open, or both closed, they simulate proto-Ross and -Weddell gyres in the subpolar Pacific and Indo-Atlantic, respectively. In all simulations, the subpolar gyres penetrate to high latitudes along the eastern boundary of the Pacific and Atlantic basins, continuing to flow westward along the Antarctic coast before



turning northward back to the mid-latitudes. In the case of the proto-Ross Gyre, subtropical waters extend along the Antarctic coast to finally merge with the EAC. These new results show that substantial heat transport from the subtropics to Antarctica is enabled by the subpolar gyres (Sauermilch et al., 2021), contrasting with Huber et al. (2004)'s results.

The opening/deepening of the Southern Ocean gateways is expected to cause changes in Southern Ocean circulation and Antarctic climate. However, it is probable that these conditions were not solely responsible for the onset of a strong ACC in the early Southern Ocean. Sijp et al. (2011) show that a deeply opened TG (1300 m) and an opened DP (1100 m) allows a strong throughflow from the Austral-Antarctic Gulf to reduce the western boundary current of the proto-Ross Sea gyre. Furthermore, under conditions when one gateway is already deep (>1,000 m), deepening of the second gateway (e.g., subsiding from 300 m to 600 m or 1500 m) results in Southern Ocean gyres that gradually weaken and shrink (Sauermilch et al., 2021). When the subpolar gyres weaken to a strength of ~10 Sv, the model forms an eastward proto-ACC. However, with late Eocene paleobathymetry (TG depth: 1500 m, DP depth: 1000 m) and paleo forcing conditions, the strength of the proto-ACC is only about 15-18% of the modern ACC's net transport (Sauermilch et al., 2021). Hill et al. (2013) also show that a strong proto-ACC (transport of >90 Sv) is established after 26 Ma. Hence, the tectonically controlled morphology of the Southern Ocean gateways before or during the E-O Transition are probably not sufficient to allow a modern-day, strong eastward flow (Hill et al., 2013; Sauermilch et al., 2021). In addition, continental separation between the South Tasman Rise and Cape Adare around 35-32 Ma means that there is suddenly an oceanic crustal pathway through the TG, which was likely at least 2500 m deep (Parsons and Sclater, 1977). This depth of the crust is enough to allow vigorous flow through the TG (Sauermilch et al., 2021). However, the marine sedimentary neodymium isotope record reveals that deep, eastward (ACC-type) flow in the TG region did not occur until ~30-29 Ma (Scher et al., 2015).

The atmospheric wind stress in the Southern Ocean and its relative latitudinal position to the gateways is likely another key factor for the development of ACC-type flow. Scher et al. (2015) compare the relative location of the Oligocene TG to the position of the polar front (boundary between polar easterly and mid-latitude westerly winds). They propose that the delay in the onset of ACC-type flow through the TG following tectonic opening is due to misalignment between the TG latitudinal opening and the westerly winds located north of the polar front. At around 30 Ma, with the migration of the northern margin of the TG into the mid-latitude westerly wind band, the alignment of westerly wind and TG throughflow allows for ACC-type flow (Scher et al., 2015).

Scher et al. (2015) therefore provide a testable hypothesis, that the inception of ACC-type flow through the TG is controlled by the relative latitudinal position of the TG and the westerly wind band. Here, we test this hypothesis in the context of different possible tectonic scenarios for Southern Ocean gateway opening. Moreover, we test whether the relative latitudinal position between the wind band and gateways has an impact on inducing the inception of strong ACC remains unknown.

## 1.2 Sensitivity of modern-day ACC to changing wind stress and momentum balance

To investigate the paleoceanographic dynamics and responses to changing wind stress in the Eocene Southern Ocean, it is first necessary to understand the dynamics of the modern-day Southern Ocean and ACC. In the present-day context, many studies have investigated the insensitivity of ACC volume transport to varying wind stress using eddy-rich models. This



phenomenon is known as "eddy-saturation" (Straub, 1993; Hallberg and Gnanadesikan, 2001; Tansley and Marshall, 2001; Munday et al., 2013). In this paradigm, increased wind stress leads to a more energetic eddy field, which is then able to transmit the increased momentum input vertically (Ward and Hogg, 2011; Marshall et al., 2017). This process can take place without steepening the mean isopycnals and therefore the mean transport does not increase.

Munday et al. (2015) conduct sensitivity tests of circumpolar transport to wind stress with an idealized eddy-permitting channel model to observe the impact of Southern Ocean gateways and bathymetric ridge on eddy saturation. The authors perform simulations using combinations of continental barriers and a submerged bathymetric ridge. The wind is fixed in position and the peak wind stress is altered. With two overlapping continental barriers and no bathymetric ridge, their model has a circumpolar transport of 93 Sv at the control wind stress, which peaks at 0.2 N m$^{-2}$. The simulated circumpolar transport is linearly dependent on the changing strength of maximum wind stress with this geometry, rising to 175 Sv with doubled wind and falling to 54 Sv with halved wind. However, with a bathymetric ridge and no continental barriers, the circumpolar transport is insensitive to wind stress, with the mean transport varying by less than 2 Sv around the control value of 94 Sv for halved/doubled wind. With both continentals barriers and bathymetric ridge, the circumpolar transport is sensitive to wind stress changes for wind stress lower than 0.2 N m$^{-2}$ (Munday et al., 2015). At the control wind stress, the transport of this geometry is 68 Sv, which rises to 79 Sv for doubled wind and falls to 50 Sv for halved wind.

Munk and Palmén (1951) first proposed a Southern Ocean momentum balance between surface wind stress and bottom form stress, with the bottom form stress being defined as a pressure difference across the abyssal topography (Olbers, 1998). Mesoscale eddies link these two stresses via eddy form stresses, which transfer momentum vertically (Johnson and Bryden, 1989; Ward and Hogg, 2011). The emergence of eddy saturation is closely linked to the momentum balance, making it critical to understanding the sensitivity of ACC zonal transport to varying wind stress. In a closed basin, like the North Atlantic, the continental barriers provide a pressure difference across the basin that balances the wind stress driving the ocean gyres. In contrast, the modern Southern Ocean does not have any continental barriers to support such a pressure gradient. However, many studies with high-resolution ocean models have investigated how the complicated submerged topography of the modern Southern Ocean can generate bottom form stress to balance the momentum input from the wind stress (Stevens and Ivchenko, 1997; Gille, 1997; Olbers et al., 2004; Munday et al., 2015; Masich et al., 2015) and confirmed the crucial role of this zonal momentum balance in eddy saturation (Marshall et al., 2017). Munday et al. (2015) associate the changes in sensitivity of circumpolar transport to wind stress with the dominant provider of form stress, which may be the continental barriers or submerged bathymetry ridge. They propose that the insensitivity of zonal volume transport occurs when bottom form stress is the primary momentum sink. This is because the mesoscale eddy field can transport momentum vertically without affecting the mean flow or mean isopycnal tilt. However, when the pressure gradient across continents dominates the sink of momentum, it reintroduces the sensitivity of circumpolar transport to changing wind stress. This is because the continents exert their pressure gradient across all depths and push back against the mean flow. Hence, the vertical eddy transport of momentum is bypassed, and the circumpolar transport is dependent upon the wind. The continental barriers and bathymetric ridge in the simulation of Munday et al. (2015) are idealized and do not represent the full complexity of the changing paleo-bathymetric conditions of the Southern Ocean during the Late Eocene. This may lead to a more complex relationship between bathymetry, circumpolar transport and zonal momentum budget than in Munday et al.'s idealized channel.





The pressure gradients (form stress) across landmasses and the pressure gradients across submerged topography constitute the total topographic form stress. Masich et al. (2015) calculate the total topographic form stress in the Southern Ocean State Estimate (SOSE) (see details in the Method 2.4) and depict its balance with zonal wind stress in the modern ACC latitudes. However, in the late Eocene (~38 Ma), continents are present at the latitudes of the TG and DP, unlike the modern Southern Ocean, and the ACC is only just forming. This may alter the momentum balance of the early Southern Ocean and potentially alter the sensitivity of the ACC transport to wind stress. As such, it is necessary to use realistic paleo-bathymetry to examine the zonal momentum balance of topographic form stress and wind stress and how it is associated with the sensitivity of circumpolar transport within the late Eocene Southern Ocean.

To investigate the impacts of wind stress, in the context of ocean gateway opening, on the early Cenozoic Southern Ocean, we use an eddy-permitting ocean model (0.25°) with realistic paleo-bathymetry for the Late Eocene (38 Ma). We conduct sensitivity experiments with different TG depths, wind stress location and strength. The details of the ocean model, paleo-bathymetry, experiment designs, and the derivation of topographic form stress are described in the Section 2. This study tests the role of relative latitudinal position between gateways and wind stress, as well as the strength of the wind stress, on the evolution of Southern Ocean gyres pattern, sea surface temperature (SST) variations and the inception of the proto-ACC, presented in the Section 3.1 and 3.2. A zonal momentum budget, to investigate the dynamics behind the late Eocene Southern Ocean circulation, is presented in the Section 3.3. In the section 4, we summarize and discuss our results.

## 2. Methods

### 2.1 Ocean model configuration

We briefly describe the ocean model configuration and paleo-bathymetry reconstruction here and refer to (Sauermilch et al., 2021) for further details. The ocean model configuration is based on an ocean-only model with no sea ice using the MIT general circulation model (MITgcm) (Marshall et al., 1997b; Marshall et al., 1997a). The model domain is circumpolar and covers the latitude range between 84°S and 25°S. The model has ¼ degree horizontal grid spacing and 50 unevenly spaced vertical levels. The model uses a nonlinear equation of state, a 7th-order advection scheme (Daru and Tenaud, 2004) and the K-profile parameterization (Large et al., 1994). Linear bottom drag is included with a coefficient of 0.0011 m/s. Surface temperature and salinity are restored to values for the late Eocene derived from the time and zonal mean of a coupled atmosphere-ocean model (Hutchinson et al., 2018). We use a 300 km sponge layer on the northern boundary of the model and set the restoring time scale to 10 days.

### 2.2 Paleo-bathymetry reconstruction

The applied bathymetry is reconstructed to 38 Ma (Hochmuth et al., 2020) for the southern part of the domain (>40°S), and extended to 25°S with the Baatsen et al. (2016) reconstruction grid. The grid is reconstructed using the plate tectonic model of Matthews et al. (2016b) in a paleomagnetic reference frame (Torsvik et al., 2008; Van Hinsbergen et al., 2015b). The southern grid is reconstructed using the sediment 'backstrippping' method by Steckler and Watts (1978). This method allows the preservation of detailed, high-resolution seafloor features and slope gradients from the present-day ETOPO (Weatherall et al., 2015) and projection to the paleo seafloor, which was not possible with previous bathymetry reconstruction methods (Müller et al., 2008; Baatsen et al., 2016). Details about the method on the bathymetric



reconstruction can be found in Hochmuth et al. (2020) and Sauermilch et al. (2020). In recent years, it has been demonstrated that seafloor slope gradients > $10^{-4}$ (m/m) have a significant impact on the subsurface eddy velocities and ocean circulation (Lacasce, 2017; Lacasce et al., 2019). The higher than typical horizontal resolution allows the model to represent these potentially important slopes with improved accuracy. Furthermore, the bathymetric

reconstruction allows us to recreate realistic continental slope regions around the Continent-Ocean-Transitions (Hochmuth et al., 2020; Sauermilch et al., 2020). The gateways depths (TG and DP) for the sensitivity tests have been manually adjusted in the paleobathymetry grids in (Sauermilch et al., 2021) with the depth values referring to the shallowest part of each gateway.

From 38 Ma to 28 Ma, the TG rapidly widened and its paleolatitude changed from ~58°S to ~53°S (Figure 1b). Meanwhile, the northward movement of DP is relatively slow, as its paleolatitude only moved to the north by 1-2 degrees from 38 Ma (about 63°S) to 28 Ma (about 61°S) (Figure 1b). In order to simulate the consequences of the northward movement of the gateways relative to the wind stress, we adjust the wind stress' paleolatitudes as the input

boundary conditions. This approach is simpler than changing the framework of the paleobathymetric reconstruction, in order to adjust the gateways' paleolatitudes. As we aim to investigate the impact of the relative latitudinal position, this approach is most constructive.

## 2.3 Experimental design

The model simulations of Sauermilch et al. (2021) with a TG/DP depth of 300 m/1000 m and 1500 m/1000 m were spun-up for 80 model years. We select the year 86 of the Sauermilch et al. (2021)'s simulation to initialize our suite of experiments. For our reference experiment, we use a revised wind stress with a maximum westerly wind of about 0.1 N m$^{-2}$ located at 53°S (Figure 1). Compared with the zonal wind stress used in the simulation of Sauermilch et al.

(2021), the revised wind forcing condition has expressed a smoother curve of polar easterlies higher than about 70°S (maximum about 0.01 N m$^{-2}$ at 74°S). For perturbation experiments, we shift the latitude of wind stress 5 degrees to the north, 5 degrees to the south, and 10 degrees to the south (presented as max 48°S, max 58°S, and max 63°S, respectively; Figure 1a). Four peak wind stress latitudes versus Southern Ocean gateways latitudes in different periods (38

Ma, 30 Ma, 28 Ma, and present) are also shown in Figure 1. Herein, we conduct eight experiments to test the sensitivity of the Southern Ocean to varied TG depths and wind stress latitudes (see details in Table 1). Additionally, we double the revised wind stress with the peak wind stress ($\tau_m$) doubled to about 0.2 N m$^{-2}$ and use the doubled wind stress in another four cases to test the sensitivity of ocean current to doubling wind stress (see details in Table 1). All

simulations are run for 60 years from the model year 86 of Sauermilch et al's simulations. Considering the adjustment period of the model due to applying the revised wind stress and shifting or doubling the revised wind stress, we will focus on the final 15 years (model years 130-145) of all the simulations to analyze results.

| Experiments | TG depths (m) | Maximum westerly wind (strength; latitude) |
|---|---|---|
| 300_max_48°S | 300 | 0.1 N m$^{-2}$; 48°S |
| 300_max_53°S | 300 | 0.1 N m$^{-2}$; 53°S |
| 300_max_58°S | 300 | 0.1 N m$^{-2}$; 58°S |
| 300_max_63°S | 300 | 0.1 N m$^{-2}$; 63°S |
| 1500_max_48°S | 1500 | 0.1 N m$^{-2}$; 48°S |
| 1500_max_53°S | 1500 | 0.1 N m$^{-2}$; 53°S |
| 1500_max_58°S | 1500 | 0.1 N m$^{-2}$; 58°S |
| 1500_max_63°S | 1500 | 0.1 N m$^{-2}$; 63°S |
| 300_max_53°S_dbw | 300 | 0.2 N m$^{-2}$; 53°S |



| 300_max_63°S_dbw | 300 | 0.2 N m⁻²; 63°S |
|---|---|---|
| 1500_max_53°S_dbw | 1500 | 0.2 N m⁻²; 53°S |
| 1500_max_63°S_dbw | 1500 | 0.2 N m⁻²; 63°S |


Table 1. Overview of sensitivity experiments. The column of "Experiments" gives names for each case, e.g., for 300_max_53°S case, 300 represents 300 m TG, max_53°S represents the maximum westerly wind of 53°S. The column of "Maximum westerly wind" provides the strength and latitudinal position of maximum westerly wind for each case.

2.4 Derivation of the zonal momentum budget and topographic form stress

The zonal momentum equation is given by:

$$\frac{\partial u}{\partial t} = \underbrace{-\xi v - \frac{\partial}{\partial x}\left\{\frac{u^2+v^2}{2}\right\} - w\frac{\partial u}{\partial z} - fv}_{(I)} \underbrace{-\frac{1}{\rho_0}\frac{\partial p}{\partial x}}_{(II)} + \underbrace{\frac{1}{\rho_0}\frac{\tau_x}{\Delta Z_s}}_{(III)} + \underbrace{\nabla\cdot\{\nu_H\nabla u\} - \frac{ru_b}{\Delta Z_b} + \frac{\partial}{\partial z}\left\{\nu_z\frac{\partial u}{\partial z}\right\}}_{(IV)} \quad (1)$$

where u is zonal velocity, $u_b$ is zonal velocity in the bottom level, $\xi$ is the vertical component
of the relative vorticity, v is meridional velocity, w is vertical velocity, f is the Coriolis parameter, $\rho_0$ is the Boussinesq reference density, p is pressure, $\tau_x$ is zonal wind stress, r is the coefficient of bottom friction, $\Delta Z_s$ is thickness of the surface level, $\Delta Z_b$ is thickness of the bottom level, $\nu_H$ is horizontal viscosity and $\nu_z$ is vertical viscosity.

Equation (1) expresses all of the individual tendency terms of the zonal momentum budget.
The left-hand side is total zonal acceleration. The first four terms (I) on the right-hand side are due to advection of horizontal momentum. The first term of term (I) is the so-called vortex force, represented by the cross product of vorticity with velocity. The second term is the zonal gradient of kinetic energy. The third term is the vertical advection of horizontal momentum, and the fourth term is the Coriolis acceleration. Term (II) is the zonal pressure gradient. Term
(III) is the zonal wind stress and inputs zonal momentum into the ocean. Term (IV) combines three terms; dissipation via horizontal viscosity, dissipation due to bottom drag and dissipation due to vertical viscosity, respectively.

We calculate the vertically integrated terms of the zonal momentum equation and estimate the average residual (the difference between total zonal acceleration and the sum of advection
(term (I) magnitude of ~10⁻²), zonal pressure gradient (term (II), magnitude of ~10⁻¹), zonal wind stress (term (III), magnitude of ~10⁻¹), and horizontal dissipation (term (IV), magnitude of ~10⁻³). The estimated residual has a magnitude of 10⁻⁶ (not shown), indicating accurate closure of the model's zonal momentum budget. Note that the depth and zonal integral of the meridional velocity are zero due to the continuity equation, so we have neglected the term of
Coriolis acceleration in the zonal momentum budget. We further zonally integrate advection, zonal wind stress, pressure gradient, and horizontal dissipation terms. These vertically and zonally integrated terms show a momentum budget for the Southern Ocean where pressure gradient and wind stress are the two dominant terms (shown in the Figure 2 top and the following equation), balancing each other:

$$\oint_x \int_{-H}^{0} \frac{1}{\rho_0}\frac{\tau_x}{\Delta Z_s}\,dz\,dx \sim -\oint_x \int_{-H}^{0} \frac{1}{\rho_0}\frac{\partial p}{\partial x}\,dz\,dx \quad (2)$$





Figure 2. Top four: the vertically and zonally integrated zonal momentum budget: total pressure gradient (total form stress) (red; Sv/s), wind stress (black; Sv/s), advection (green; Sv/s), horizontal dissipation (magenta; Sv/s), vertical dissipation (light blue; Sv/s) in the
1500_max_53°S, 1500_max_63°S, 1500_max_53°S_dbw and 1500_max_63°S_dbw cases. Bottom four: the zonally, vertically integrated of momentum budget between wind stress (black; Sv/s), total topographic form stress (blue; Sv/s) and total pressure gradient (red; Sv/s) in the 1500_max_53°S, 1500_max_63°S, 1500_max_53°S_dbw and 1500_max_63°S_dbw cases.

In this zonally and vertically integrated momentum balance, the pressure gradient field can be recognized as the pressure gradient across topography (water leaning on land) rather than the pressure gradients in the ocean interior (water leaning on water) (Masich et al., 2015). The depth-integrated total zonal pressure gradient can be further decomposed into three components via Leibniz' rule:

$$\int_{-H}^{0} -\frac{1}{\rho_0}\frac{\partial p}{\partial x}\,dz = \underbrace{-\frac{1}{\rho_0}\frac{\partial}{\partial x}\int_{-H}^{0} P\,dz}_{\{\alpha\}} + \underbrace{\frac{1}{\rho_0}P_{(z=0)}\frac{\partial z}{\partial x}}_{\{\beta\}} + \underbrace{\frac{1}{\rho_0}P_{(-H)}\frac{\partial H}{\partial x}}_{\{\gamma\}} \quad (3)$$

where $P_{(z=0)}$ is the atmospheric pressure at the surface, and $P_{(z=-H)}$ is the pressure in the bottom layer and H is the ocean depth. $\alpha$ is the transfer of zonal momentum from continental boundaries to the ocean (Munday et al., 2015), $\beta$ is the transfer of zonal momentum from the
atmosphere to the ocean (Masich et al., 2015), and $\gamma$ is the bottom form stress, which is the transfer of zonal momentum from submerged bathymetric features to the ocean (Munday et al., 2015). We can assume that the atmospheric pressure is zero, so $\beta$ can be neglected. Hence, the depth integrated total pressure gradient can be reduced to the sum of pressure gradients across continents and submerged topography ($\alpha + \gamma$), which is the total topographic form stress.

Following Masich et al. (2015), the zonally vertically integrated total zonal pressure gradient (or total topographic form stress) can be discretized as:

$$-\oint_{x}\int_{-H}^{\eta}\frac{1}{\rho_0}\frac{\partial p}{\partial x}\,dz\,dx = \sum_{x}\sum_{-H}^{\eta}\frac{1}{\rho_0}\Delta P_t\Delta z \quad (4)$$

Where $\Delta P_t$ is the pressure difference across all topographic features (continental barriers and submerged bathymetric features) in our model configuration. It represents the difference
between the pressure of fluid cells adjacent to the eastern face of continents or submerged bathymetric features and the pressure of fluid cells adjacent to the western face of the same (eastern minus western). $\Delta z$ is the height of the model cell. Through the discretization in the equation 4, we can simply extract the total topographic form stress from the zonally vertically integrated total zonal pressure gradient field. More detail on the calculation of topographic
form stress, and errors associated with the use of partial model cells, can be found in the method section of Masich et al. (2015) and Supplementary Information section 1 and 2.

### 3   Results

3.1 Sensitivity of Southern Ocean gyres and sea surface temperature to changing TG depth and
wind stress latitude.





Before the inception of the ACC, the early Eocene Southern Ocean basins were dominated by wind-driven gyres, anti-clockwise for the subtropical gyres, and clockwise for the subpolar gyres (Huber et al., 2004; Huber and Nof, 2006; Hill et al., 2013). We reproduce this oceanic

pattern, as shown in Figure 3c. When the TG is 300 m deep, migrations of the wind band adjust the position of the ocean gyres boundary and slightly alter the gyres' spatial scale. For example, when the maximum westerly wind peaks at 53°S, the boundary between subtropical and subpolar gyres, indicated by the red line in Figure 3c, is also positioned at about 53°S (We extract the latitudinal position of red lines according to the averaged latitude of the streamline

with zero meridional velocity, along which the ocean current only flows zonally). The positional alignment of maximum westerly wind and gyres boundary is consistent with Sverdrup theory (Sverdrup, 1947). As the latitude of the maximum westerly wind is shifted to 48°S, 58°S, and 63°S, the gyre's boundary accordingly varies its latitudinal position to 48°S, 58°S, and 63°S (Figure 3). Meanwhile, the spatial scale of the subpolar gyre (green) becomes

about 30% smaller for a wind shift from 48°S to 63°S (Figure 3a, c, e, g).

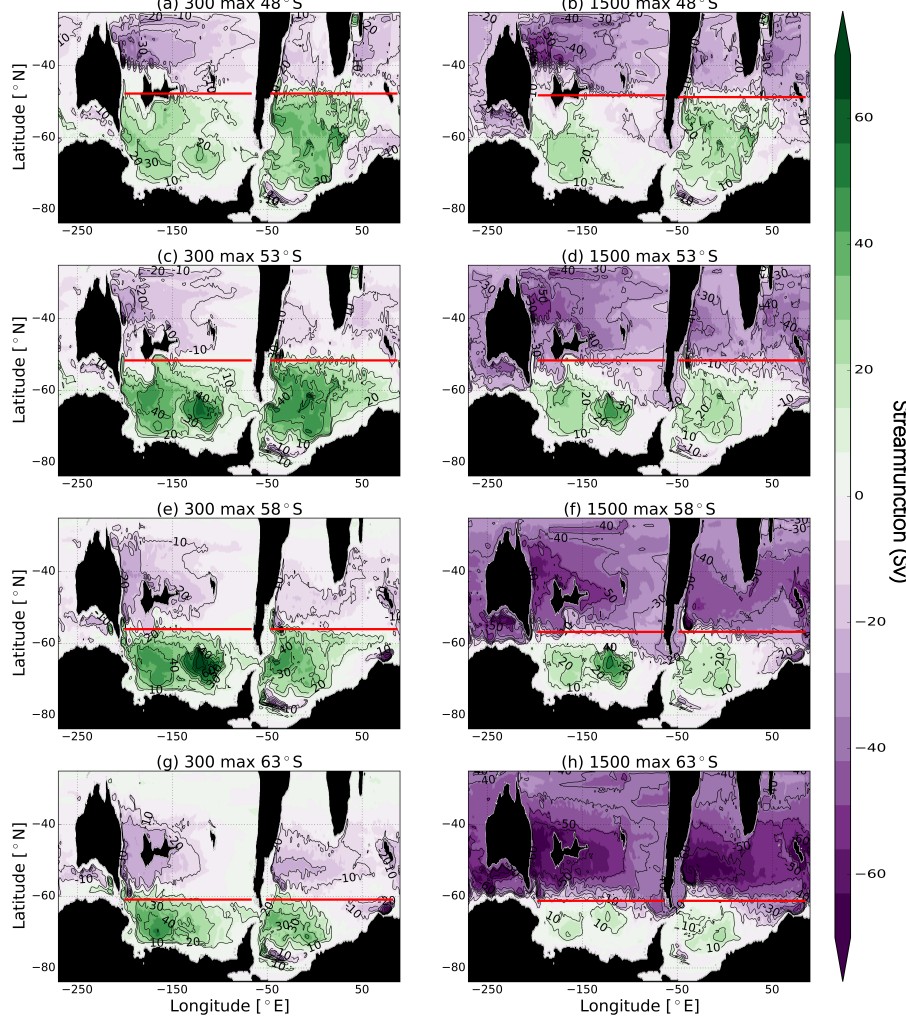



Figure 3. The 15-model year time-average Southern Ocean circulation patterns (annual mean and depth integrated stream function) under stepwise deepening of the Tasman Gateway from
300 m (left) to 1500 m (right) depth and wind stress meridional migrations (from top to bottom for both two rows). Purple colors indicate counterclockwise gyres and green colors show clockwise gyres. Red lines indicate the theoretical gyre boundary between counterclockwise and clockwise gyres which is strictly aligned with the maximum westly wind $\tau_m$. The latitudes of $\tau_m$ applied in four cases are 48°S (a, b), 53°S (c, d), 58°S (e, f), and 63°S (g, h), respectively.


In the simulations of Sauermilch et al. (2021), deepening of the TG to 1500 m results in the subtropical gyres at latitudes of 40°S-60°S growing significantly and dominating the mid-latitudes of the Southern Ocean. In contrast, the subpolar gyres in both the Pacific and Indo-Atlantic sectors weaken and shrink. An eastward circumpolar current, with a transport of about
20 Sv, flows through DP (Sauermilch et al., 2021). Here our simulations indicate that southward migrations of the wind stress lead to further shrinking of the subpolar gyres (Figure 3f, h). This change is most remarkable in the 1500_max_63°S case (meaning the case with 1500 m TG and maximum westerly wind located at 63°S, see Table 1), where the northernmost latitude of the subpolar gyres has been restricted to higher than 63°S. The subpolar gyres shrink
over 50% in spatial area and the maximum transport of the subpolar gyres decreases to 10 Sv, as shown by the contours.

The changes in gyre structure due to wind shifts also induce significant changes in the modelled sea surface temperatures. When the TG is 300 m deep, Sauermilch et al. (2021) show that the
colder high-latitudinal waters (about 10-12°C) in the centers of the subpolar Pacific and Atlantic gyres are enclosed by warmer subtropical surface waters (about 15-17°C). The subpolar clockwise gyres transport the subtropical waters to the Antarctic coast. We also present similar temperature patterns with mean sea surface temperatures at high latitudes (higher than 60°S) of about 14 °C with the shallow TG and maximum westerly wind at 53°S
(Figure 4c). The sea surface temperatures under the shallow TG condition are sensitive to further southwards shifts of the westerly wind, as shown in Figure 4e, g. As the westerly wind moves southward, the warm subtropical water gradually dominates the coastline of Antarctica, and the cold water occupies a smaller and smaller area at the center of the subpolar gyres. Especially in the model where the TG is 300 m deep and the maximum wind stress is at 63°S
(Figure 4g), the cold waters in the Pacific shrink by over 50% in spatial area. The cold water almost disappears in the South Atlantic and the mean sea surface temperature at high latitudes increases to about 16°C.

When the TG is 1500 m deep, the subtropical waters are still able to reach the Antarctic coast
and encircle the colder water in the center of the clockwise gyres when the maximum wind stress is at 48°S and 53°S. The sea surface temperatures along the Antarctic coast in these two cases are around 16°C (Figure 4b, d). Therefore, our simulation supports the argument of Sauermilch et al. (2021), who concluded that Antarctica could have been warmed by subtropical waters prior to the E-O transition. However, as the wind stress shifts farther
southward (63°S), the result is quite different compared to models with a shallow TG. The warm waters from low latitudes gradually fail to reach the polar region, as indicated by the current velocity vectors (Figure 4f, h). In the model where the TG is 1500 m deep and the maximum wind stress at 63°S (Figure 4h), the absolute sea surface temperatures along the entire Antarctic coast are lower than 12°C. This leads to the formation of a large meridional
temperature gradient across the Southern Ocean (~7°C temperature difference from New Zealand to the Antarctic coast, see Figure S4). This current does not flow poleward to form





closed gyres, as seen from the streamfunction in Figure 3h, and thermally isolates the Antarctic
continent.

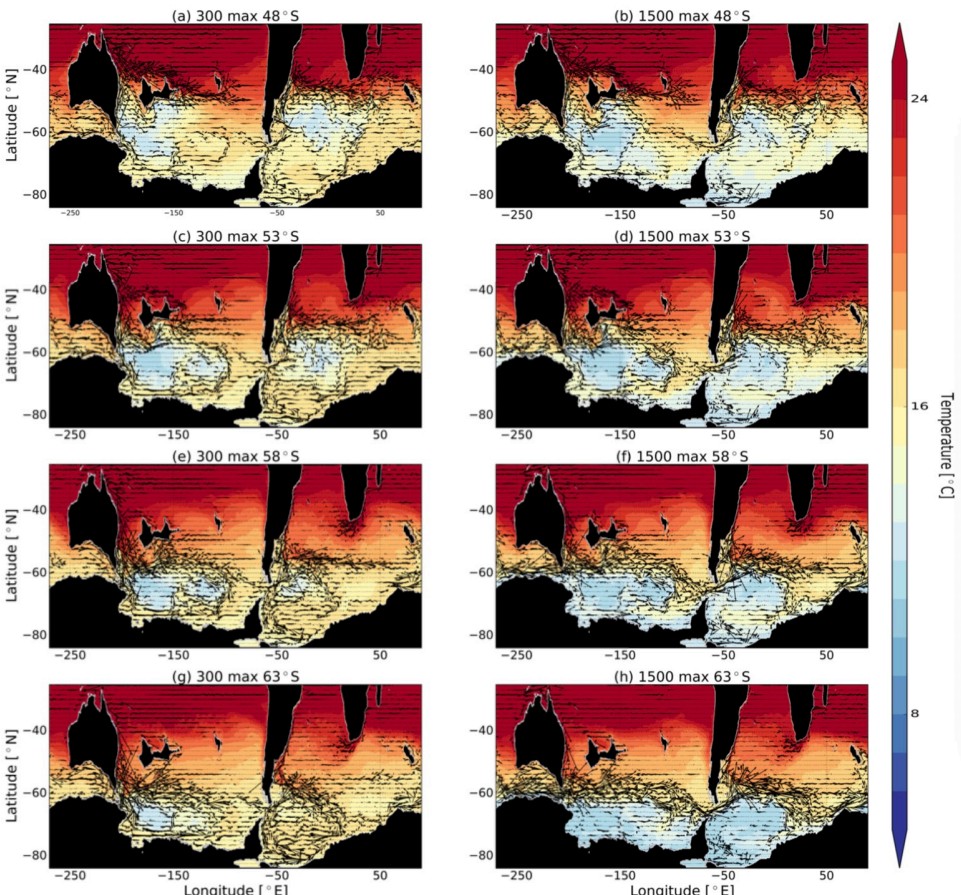

Figure 4. Annual mean sea surface temperatures (100 m depth) of the Southern Ocean under
the deepening of the Tasman Gateway from 300 m (left) to 1500 m (right) depth and wind
stress shifts (from top to bottom). The overlaid arrows demonstrate local current velocities.

3.2 Response of throughflow transport to doubling wind stress

The maximum zonal volume transport of our proto-ACC does not exceed 50 Sv (TG
throughflow transport: 47.9 Sv, DP throughflow transport: 38.3 Sv; Table 2, 1500_max_63°S),
which is about a third of the modern-strength ACC (transport of 137 Sv; Meredith et al. (2011)).
In the traditional paradigm, wind stress acts as an important driver of ocean current. Thus,
increased wind stress is expected to strengthen the input of zonal momentum to the ocean and
accelerate ocean currents. Here, we simulate the consequences of doubling the wind stress
(from 0.1 to 0.2 N m$^{-2}$) with different gateway depths (TG at 300 m, 1500 m) and different
latitudinal positions of maximum wind stress (at 53°S and 63°S). The variations of the TG and
DP transport are shown in Figure 5. The total TG transport is strengthened in all four
simulations due to the doubled westerly wind stress. When TG is 300 m, TG transport increases
43% and 72% with peak wind stress located at 53°S and 63°S, respectively. When TG is 1500





m, TG transport is increased by 60% and 57% with peak wind stress located at 53°S and 63°S, respectively. The effect is particularly strong in the model where the TG depth is 1500 m and maximum wind strength at 63°S where the total TG throughflow transport is enhanced to about 75.4 Sv. Conversely, the total DP throughflow transport is not strongly influenced by the changes in wind stress with the increases in most simulations lower than 35%.

| Shallow TG | 300_max_48°S | 300_max_53°S | 300_max_53°S_dbw | 300_max_58°S | 300_max_63°S | 300_max_63°S_dbw |
|---|---|---|---|---|---|---|
| TG (Sv) | 2.3 | 2.8 | 4.0 | 3.9 | 3.9 | 6.7 |
| DP (Sv) | -4.0 | -7.3 | -11.9 | -8.0 | -7.9 | -8.1 |
| Deep TG | 1500_max_48°S | 1500_max_53°S | 1500_max_53°S_dbw | 1500_max_58°S | 1500_max_63°S | 1500_max_63°S_dbw |
| TG (Sv) | 18.5 | 29.3 | 46.9 | 41.4 | 47.9 | 75.4 |
| DP (Sv) | 7.5 | 13.8 | 18.5 | 21.7 | 38.3 | 44.8 |


Table 2. Net TG and DP throughflow volume transport (Sv). Positive values indicate eastward transport, negative values present westward transport. All values are the annual mean transport of the final 15 years of the simulations.

Dynamically, in a homogenous ocean, currents are strongly constrained by f/H contours
(Johnson and Hill, 1975). Even in the presence of stratification, the submerged topography of the Southern Ocean can block f/H contour, which steers the current to maintain conservation of potential vorticity (Marshall, 1995). The blocking of f/H contours reduces the velocity below the bathymetric level and allows the transport due to thermal wind shear to dominate the ocean current transport (Munday et al., 2015). Given the complex paleo-bathymetry applied in this
model study, we decompose the total TG/DP throughflow transport into two components contributed by bottom flow transport $T_b$ (transport below the bathymetric level) and thermal wind transport $T_{tw}$ (transport above the bathymetric level) using three equations:

$$U_b = \frac{1}{H_b} \int_{-H}^{-H_b} U \, dz \quad (6)$$

$$T_b = H_b \int U_b \, dy \quad (7)$$

$$T_{tw} = T_{total} - T_b \quad (8)$$

Equation 6 calculates the zonal bottom flow velocity $U_b$. By looking at hydrographic sections of the local bathymetry and zonal velocity (see the example of TG in the Supplementary Information 5 and Figure S5) for the TG and DP, respectively, we select a special model level for both TG and DP below which the current velocities are nearly homogenous, and above the
level the velocities show strong horizontal and vertical gradients. Then we use a vertical average of zonal velocity below that model level as the $U_b$. In the shallow TG cases, e.g. 300_max_53°S case, we choose model level 30 (~446 m) for the TG. In the deep TG cases, e.g. 1500_max_53°S case, we select the model level 32 (~668 m) for the TG. In all cases, we select the model level 33 (~728 m) for the DP. Equations 7 and 8 calculate the bottom flow
transport and thermal wind transport, which are referred to equations (4) and (5) in Munday et al. (2015).





With a 300 m TG (Figure 5 a,b), $T_{tw}$ dominates the entire TG throughflow transport, while the DP throughflow transport is dominated by a westward bottom flow (negative $T_b$). Both $T_{tw}$ and $T_b$ for each gateway are small in the case of a 300 m TG (<4 Sv for TG and <9 Sv westward for DP). The response of $T_b$ and $T_{tw}$ to doubling wind stress are also shown in Figure 5. Both latitudinal shift and doubling of the wind stress cannot increase the TG transport ($T_{total}$, $T_{tw}$, and $T_b$) to larger than 7 Sv and the DP westward transport ($T_{total}$, $T_{tw}$, and $T_b$) to larger than 12 Sv, with 300 m TG.

When the TG deepens to 1500 m (Figure 5c and d), the deeper bathymetry allows for the generation of a strong bottom flow ($T_b$) through the TG. Doubling wind stress strengthens the $T_b$ through the TG significantly (89% and 77% increase when peak wind stress locates at 53°S and 63°S), while changes in $T_{tw}$ through the TG due to doubled wind stress are small (16% and 18% when peak wind stress locates at 53°S and 63°S).

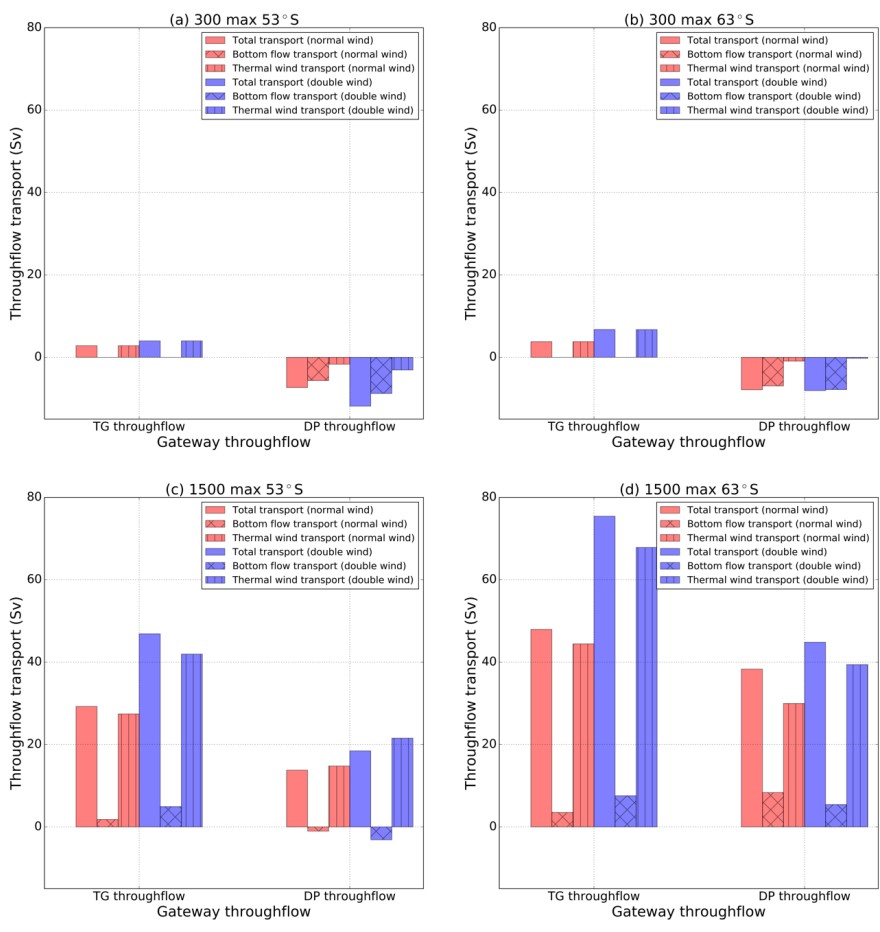

Figure 5. The contrast of TG (red) and DP (blue) throughflow volume transport in normal-strength and double-strength wind stress experiments. The total transport is decomposed into bottom flow transport and thermal wind transport.



With the deeper TG, a big variation occurs in the dominance of DP throughflow transport, changing from $T_b$ to $T_{tw}$. Both components of DP throughflow transport are less sensitive to the doubling wind stress (21% and 15% increase for $T_b$ and $T_{tw}$) with the peak wind stress located at 63°S. In addition, two points are noticeable: (1) The $T_b$ through the DP is still negative in the 1500_max_53°S case, which means that there still exists the westbound bottom flow through the DP with a deep TG and maximum westerly wind located at 53°S. (2) In the 1500_max_63°S case, the DP bottom flow is now eastward, but its transport weakens with doubling wind stress.

3.3 Zonal momentum balance and topographic form stress in the Late Eocene Southern Ocean

The shallower and narrower gateways of the Eocene may impact the momentum balance and topographic form stress by allowing the continents to play a larger role in balancing wind stress. We have diagnosed a zonally and vertically integrated zonal momentum balance for the Eocene Southern Ocean, in which zonal momentum input by the wind is balanced by the zonal pressure gradient (Figure 2 top). As wind stress shifts 10 degrees to south and doubles its strength, the vertically and zonally integrated pressure gradient also shifts to the south and enhances. The maximum acceleration due to the zonally integrated wind stress in the 1500_max_63°S and 1500_max_63°S_dbw cases is smaller than 1500_max_53°S and 1500_max_53°S_dbw. This is due to the zonal distance around the Earth which is smaller at higher latitudes.

In our simulations, the blue curve (total topographic form stress) and red curve (total pressure gradient) are almost coincident (Figure 2 bottom). The total topographic form stress is calculated from pressure differences across topographic features using the method of Masich et al. (2015). The total pressure gradient is the zonal and depth integral of term (II) in equation (1). This indicates the accuracy of Masich et al. (2015)'s method of estimating the total topographic form stress. The slight difference between calculations is mostly due to ambiguity over the pressure and depth of partial cells (see Supplementary Information 2 and Figure S2).

Previous studies have described different regimes in the topographic form stress for the modern Southern Ocean at different depths (Gille, 1997; Grezio et al., 2005; Masich et al., 2015). For example, Masich et al. (2015) divided the integrated topographic form stress into two regimes for modern Southern Ocean: shallower/deeper than 3700 m. From the surface to 3700 m, the integrated topographic form stress can balance with wind stress, while below 3700 m, the topographic form stress signal has a large-scale horizontal structure but cancels over the whole basin in the zonal integral (Masich et al., 2015). Nevertheless, the different bathymetry of the Late Eocene Southern Ocean may change this picture.

Figure 6 shows the comparison between the horizontally and zonally integrated topographic form stress, vertically integrated to every level, and integrated wind stress. Generally, the depth integrated topographic form stress contributions in the four cases have similar variation. Near the sea surface, the topographic form stress signal is positive and reaches a maximum value at a shallow depth ($H_s$, e.g. 380 m in the 1500_max_53°S). The surface or upper layer of the Southern Ocean has few submerged bathymetric features such that the pressure gradient across continents must be the primary source of topographic form stress. In these upper layers the continents are pushing in the same direction as the wind stress. Below the near-surface, the signal decreases in magnitude until a mid-depth ($H_m$, e.g. 610 m in the 1500_max_53°S) where the integrated topographic form stress is zero. In this layer, the initial eastward push from the topographic form stress has been matched by a corresponding westward push, such that they cancel each other out at $H_m$. At this depth the continents exert zero net push on the ocean.



Finally, below $H_m$, the depth-integrated topographic form stress is increasingly negative to the ocean floor. This deep layer is where the bathymetry pushes back and balances the wind. At deep depth ($H_d$, e.g. 3880 m in the 1500_max_53°S), the integrated topographic form stress is of sufficient magnitude to balance the wind stress. Compared with the reconstructed

paleobathymetry (Figure S3) used in all simulations, we can find that $H_s$ probably corresponds to continental shelves in some regions, like North Australia, South America, and Antarctica. $H_m$ in the cases of 53°S is probably controlled by the tops of seamounts in the north of New Zealand and southeast of Africa while $H_m$ in the cases of 63°S may be associated with the top of mid-ocean-ridge in the circumpolar belt. Finally, $H_d$ may correspond to subduction zone of

the mid-ocean-ridge.

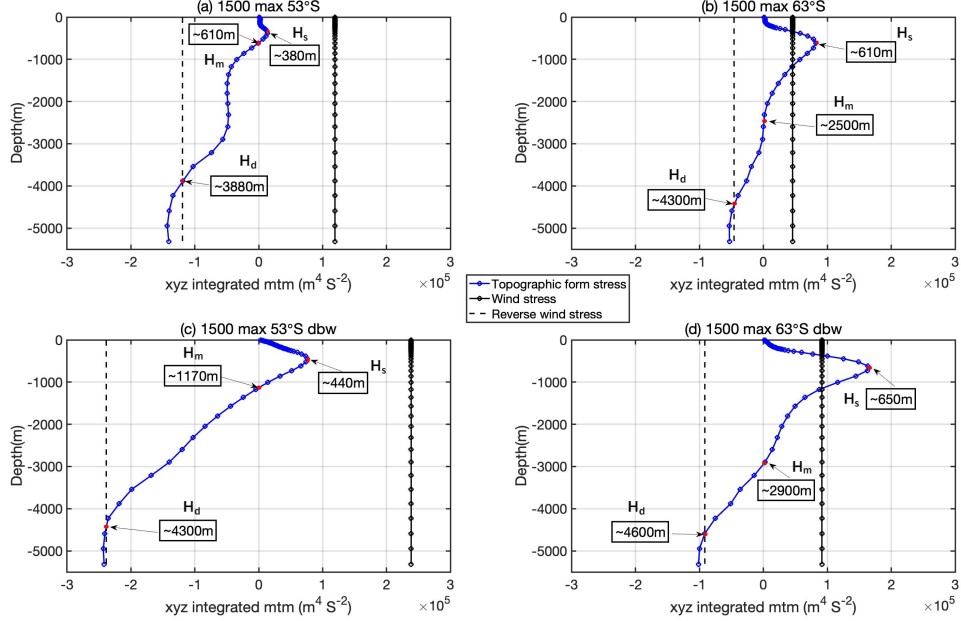

Figure 6. Blue curve: topographic form stress integrated from surface to various depths. Black line: wind stress integrated from surface to various depths. Dashed line: reversed wind stress

integrated from surface to various depths (same strength but opposite sign with depth integrated wind stress). There are three unique depths indicated by red points in all four cases. The shallow depth ($H_s$) is the first local maximum in the depth integrated topographic form stress, which has the same sign as the integrated wind stress (e.g. ~380 m in the subplot a). The mid-depth ($H_m$) is where depth integrated topographic form stress is zero (e.g. ~610 m in the subplot a).

The deep depth ($H_d$) is where depth integrated topographic form stress intersects with reverse wind stress or has the same contribution but opposite sign with depth integrated wind stress (e.g. ~3880 m in the subplot a).

Dividing the ocean into latitude bands can highlight changes in the depth integrated

topographic form stress signal (Masich et al., 2015). We divide our model domain into three latitude bands by considering the ocean circulation and bathymetry. These bands are the northern region (25°S~50°S), the circumpolar belt (50°S~70°S), and the polar region (70°S~83°S). These three regions have different surface ocean areas due to the change in longitudinal extent of the Earth with latitude and the position of the continents. As such, we



normalize the depth integrated topographic form stress and wind stress by surface ocean area in the three regions.

From Figure 7, we can see that the northern region in all cases has a positive depth integrated topographic form stress signal near the ocean surface, as with the full domain signal in Figure

6. This signal reaches a maximum around $H_s$. This suggests that it is the subtropical gyres of the northern region responsible for these shallow maxima. Below $H_s$, the pressure gradient across submerged topography acts to cancel the positive signal. When the peak westly wind is at 53°S, the wind stress in the northern region is positive (westerly wind dominates). As the peak westerly wind shifts from 53°S to 63°S, the wind stress contribution in the northern region

changes sign from positive to negative. The gyres are still able to form because they depend upon the curl of the wind, rather than its direction (Stommel, 1948; Munk, 1950). However, the vertical structure of the topographic form stress is characteristically different to the 53°S wind stress cases because it does not need to become negative at depth to balance a westward wind stress. In the meantime, the subtropical easterlies dominate the northern region, and the

subtropical gyre expands in size.

## 1500m TG, normal wind stress (~0.1 N m$^{-2}$)

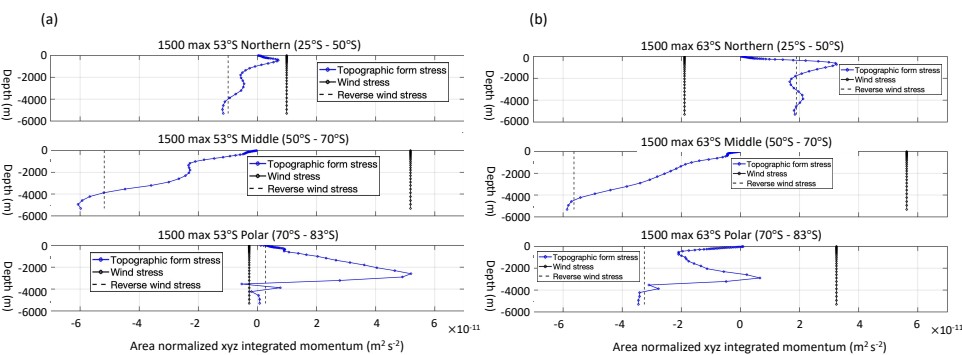

## 1500m TG, doubled wind stress (~0.2 N m$^{-2}$)

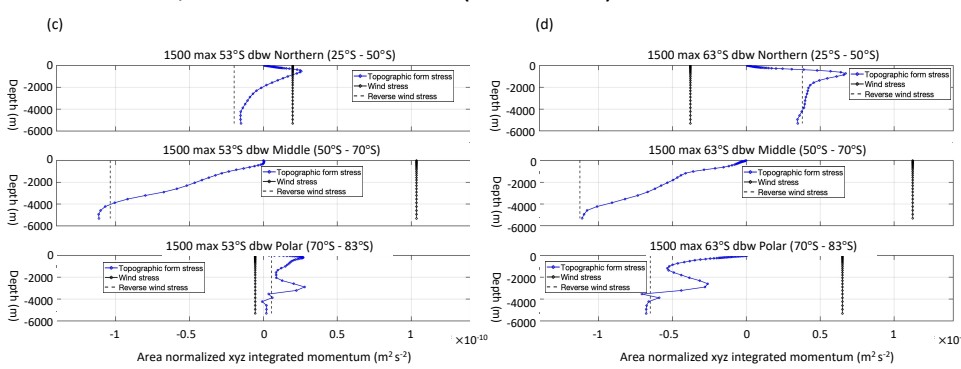

Figure 7. Area normalized depth integrated topographic form stress (Blue curve) and wind stress (Black line) in three channels (25°S ~50°S, 50°S ~70°S, 70°S ~83°S). Dashed line: reversed wind stress integrated from surface to various depths (same strength but opposite sign

with depth integrated wind stress).



In the circumpolar belt, all panels show a consistently negative depth integrated topographic form stress. This acts to balance the eastward wind stress at $H_d$. This is the signal of circumpolar flow with the surface wind stress balanced by bottom form stress, via vertical flux of momentum by mesoscale eddies, that prevails in the modern Southern Ocean (Johnson and Bryden, 1989; Ward and Hogg, 2011). It is this region that is dominating the whole ocean balances, in Figure 7, at depth. All four experiments have circumpolar flow in excess of ~30 Sv (Figure 5).

In the polar region, we find a positive topographic form stress signal above ~3800 m in the 53°S cases. This is balancing a net westward wind stress and has a similar vertical structure to that noted for the northern region above. This may indicate a characteristic contribution of the subpolar gyre. The positive topographic form stress signal is not present in the 63°S cases. The change in structure of the wind results in a net eastward wind stress in these experiments. As such, the topographic form stress must become negative. However, the vertical structure has a tendency towards more negative values at depth, rather than variation around the value that balances the wind stress seen in Figures 7b (upper panel) and 7c (lower panel). This is probably induced by the shrinkage of the subpolar gyre and the inception of the proto-ACC, which intrudes into this latitude band and influences its momentum balance.

Almost all submerged topographic features in our model have bathymetry deeper than $H_s$ (see the reconstructed paleo-bathymetry in the Figure S3). Hence, $H_s$ can be recognized as a contribution-dividing depth, above which the topographic form stress is the contribution largely sustained by the pressure gradient across continents, and below which the topographic form stress is contributed by submerged topography and the rest of continents. We apply $H_s$, which is allowed to vary between experiments, to decompose the total topographic form stress into shallow (shallow TFS) and deep topographic form stress (deep TFS). The model level $H_s$ is applied in the equation (5) (in the section 2.5 of Method) to give shallow/deep topographic form stress contributions:

Shallow topographic form stress: $\sum_x \sum_{-H_s}^0 \frac{1}{\rho_0} \Delta P_t \Delta z$      (6)

Deep topographic form stress: $\sum_x \sum_{-H}^{-H_s} \frac{1}{\rho_0} \Delta P_t \Delta z$        (7)

Figure 8 shows that the subtropical easterly wind stress is mostly balanced by the shallow TFS contribution (positive sign). Comparing with the results shown in Figures 6 and 7, the positive signal of the depth integrated topographic form stress in the upper regime (e.g. above 610 m in Figure 6d) and in the northern region is mostly contributed by shallow TFS. This is generated by the subtropical gyres in the surface layer to cancel the momentum input from subtropical easterly wind stress (Figure 8d). All panels of Figure 8 also show that the westerly wind stress is largely balanced by deep TFS (negative sign), which is caused by the eastward current (e.g. proto-ACC) flowing across submerged topography and mostly contributes to the negative signal of the depth integrated topographic form stress in the circumpolar belt.

In the 1500_max_53°S case (Figure 8a), the deep TFS has some positive signals to balance the easterly wind stress in the latitudes of 25°S to 29°S. In comparison, in the 1500_max_63°S case, the deep TFS at these latitudes is negative (Figure 8b). The maximum positive signal of the depth integrated topographic form stress in the 1500_max_63°S case is larger than depth integrated wind stress (Figure 7b). This is due to the large shallow TFS contribution in the latitudes of the northern region, which is then balanced with both wind stress and deep TFS. A



weak positive signal of shallow TFS occurs in the latitudes of 60°S to 75°S in the 1500_max_53°S case but disappears in the 1500_max_63°S case (Figure 8a and 8b), which may be related to the inception of the proto-ACC. With doubled strength wind conditions (0.2 N m$^{-2}$; Figure 8c and d), the sensitivity of the shallow TFS to doubled wind is strong in the latitudes of subtropical easterlies while it is weak in the latitudes of westerlies. The deep TFS shows the constantly strong sensitivity in all latitudes. The sensitivity or insensitivity of the shallow/deep TFS is associated with the responses of TG and DP transport to doubled wind stress, which will be discussed in the following section.

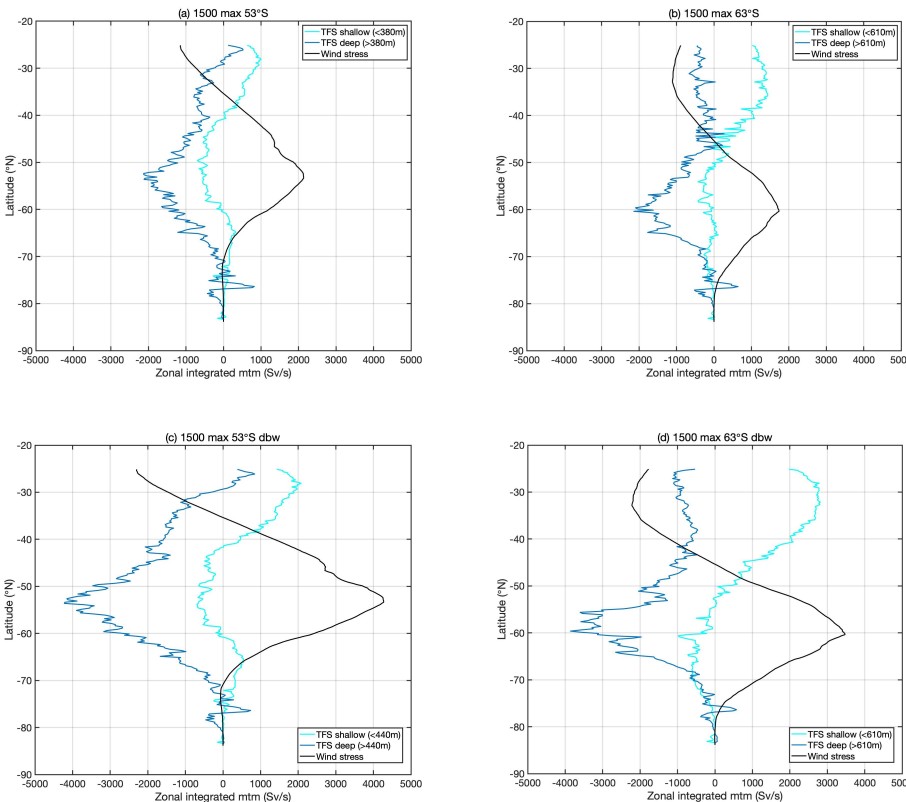

Figure 8. The zonally, vertically integrated of momentum budget between wind stress (black; Sv/s), shallow topographic form stress (blue; Sv/s) and deep topographic form stress (dark blue; Sv/s) in the 1500_max_53°S, 1500_max_63°S, 1500_max_53°S_dbw and 1500_max_63°S_dbw cases.

## 4   Discussion

Early studies have suggested that the E-O Southern Ocean circulation and climate were largely influenced by the opening/deepening of oceanic gateways (Barker and Burrell, 1977; Kennett, 1977; Barker and Thomas, 2004; Exon et al., 2004). Nevertheless, other studies, using proxy evidence, point out an inconsistency in the timing of the TG opening and Antarctic glaciation and propose that the opening of the TG is probably not the main cause of the Antarctic cooling (Huber et al., 2004; Stickley et al., 2004; Wei, 2004). It has also been suggested that the decline



of atmospheric $CO_2$ might be the primary trigger of the E-O transition (Anagnostou et al., 2016; Foster et al., 2017). A more recent model study conducted by Sauermilch et al. (2021), using an eddy-permitting ocean simulation with realistic paleobathymetry, shows that the deepening of gateways leads to a weakening of subpolar gyres, and subsequently a cooling of Antarctic
surface waters. This change in understanding, relative to previous studies, is due to the use of an eddy-permitting ocean model combined with a realistic paleobathymetry, allowing for a more accurate representation of ocean circulation.

Most past studies only considered the gateways as either opened or closed rather than
understanding the role of progressively increasing gateways' depths (e.g. Sijp et al., 2011, Hill et al., 2013). In addition, changes in TG and DP depth are rarely considered simultaneously. The alignment of westerly winds with the TG has also been proposed to be a trigger of proto-ACC (Scher et al., 2015), but this hypothesis has never been tested in a model. In this study, we use an eddy-permitting ocean model with realistic paleo-bathymetry to show that gateway
depth is not the only factor influencing the circulation of the Eocene Southern Ocean. The strength and relative latitudinal position of westerly winds also play a significant role in establishing a proto-ACC. In particular, we address the sensitivity of ACC strength, subpolar gyre strength, and SST distribution in the Southern Ocean, to wind stress position and strength.

**4.1 Consequences of a latitudinal change of the wind stress maximum**

Our simulations indicate that a deep TG (1500 m) and southward shifts in wind stress lead to the shrinking of subpolar gyres (Figure 3) and the cooling of the Antarctic surface waters (Figure 4). Wind stress and continental barriers sustain the large-scale gyres in the oceanic
basins (Munk, 1950) and the latitudinal position of gyres boundaries is aligned with the position of maximum wind stress (Sverdrup, 1947). Hence the southward movements of wind stress can narrow the spatial scale of subpolar gyres under the restriction of the Antarctic continent, while northward wind shifts extend the subpolar gyres' spatial scale. This change in gyre circulation, associated with a southward shift of westerly winds, agrees with seismic evidence
and Neodymium records from offshore New Zealand, which show a transition from southern-sourced waters to northern-sourced waters at 36 Ma (Sarkar et al., 2019), even though the authors have attributed this change to an onset of a proto-ACC.

**4.2 The role of the TG deepening and wind stress location on the development of the proto-
ACC**

Our results show that with a shallow TG (300 m), the maximum TG transport is only 3.9 Sv with the maximum wind stress at 63°S (Table 2). This corresponds to when the wind stress maximum is aligned with the TG. Meanwhile the opening of the DP (1000 m) allows a negative
net DP transport, leading to a westward flow from the Atlantic to the Pacific (Table 2). This suggests that a strong proto-ACC may not be possible while the TG is shallow. Some studies use numerical models (Sijp et al., 2011) or proxies (Stickley et al., 2004) to infer that a deep TG allows a strong throughflow as part of the formation of the proto-ACC. The model of Sijp et al. (2011) has a 1300 m TG and 1100 m DP, which results in a 66 Sv circumpolar flow
replacing the subpolar gyres throughout the Southern Ocean. However, our 1500_max_53°S case (1500 m TG and 1000 m DP) simulates a 29.3 Sv TG transport and a 13.8 Sv DP transport. In addition, two subpolar gyres are still vigorous (maximum transport streamfunction ~40 Sv) in this case. Hence, the TG deepening can induce an eastward TG throughflow as an initial phase of a proto-ACC. However, another process may be required for the development of a





strong proto-ACC that is capable of contributing to the observed changes to the E-O Southern Ocean circulation and climate.

In addition to the impact of TG deepening on the onset of the proto-ACC, we also consider the role of the meridional position of the wind stress maximum. Scher et al. (2015) propose the
inception of the proto-ACC was triggered by alignment of the TG with the westerly wind band. In our 1500 max 58°S case, the northern margin of the TG is in the latitudes of the westerly wind band (Figure 1) and the TG transport increases from 29.3 Sv of our 1500_max_53°S case to 41.4 Sv (Table 2). However, subtropical water is still transported to the Antarctic coast by the subpolar gyres and the DP transport is only 21.7 Sv. Our simulation is therefore consistent
with the hypothesis of Scher et al. (2015) who hypothesized that a combination of the alignment of winds, together with gateway deepening, led to an increased transport through the TG. Nevertheless, the work of Scher et al. (2015) does not address the role of DP deepening.

Our model shows a stronger proto-ACC (TG throughflow transport ~47.9 Sv, DP throughflow
transport ~38.3 Sv; Table 2) in the 1500_max_63°S case, in which the maximum westerly wind stress is aligned with both the TG and the DP (Figure 4h). With a 1500 m TG, the strong TG throughflow invades the subpolar Pacific sector, forming a large-scale eastward ocean current broadly along the latitude of the maximum westerly wind. This current extends downstream towards the DP. Once the eastward current reaches around the tip of South America, its
following pathway is quite distinct in different wind stress cases (Figure 4d, f, h). For instance, when the maximum westerly wind is at 53°S or 58°S, the continental barrier forces the eastward current to flow poleward, then mostly reform into subpolar gyres (Figure 4d, f). However, if carefully comparing the latitudes of different wind stress and both gateways at 38 Ma (Figure 1), we find that the maximum westerly wind of 63°S is at the southern margin of the TG and
northern margin of the DP. This condition enables the eastward current to mostly penetrate the DP without turning poleward to replenish the subpolar gyres (Figure 4h). Thus, the latitudinal alignment of deep gateways (TG and DP) and the maximum westerly wind may be a prerequisite for the inception of a strong proto-ACC and the thermal isolation of the Antarctic.

As noted earlier, the TG moved northward by ~5° and DP by ~1-2° between 38 and 28 Ma (Figure 1). As our simulations are ocean-only, it is difficult to estimate the behavior of the westerly wind position itself, which is also influenced by Antarctic cooling across the E-O (~ 34 Ma). Very little information exists about the impact of Antarctic cooling, especially Antarctic ice sheet (AIS) expansion, on the latitudinal position of the westerly wind. However,
results from an E-O coupled atmosphere-ocean model indicate that the westerly wind shifts nearly 10° southward, with the appearance of AIS (Kennedy et al., 2015). Assuming a westerly wind (peak wind stress located at 53°S) shift southward by 10° at ~34 Ma, and TG/DP moves northward by ~5°/1-2°, respectively, around the same period (from 38 to 28 Ma), it is likely that around 28 Ma, the gateways and wind maximum show the best latitudinal alignment.

Our bathymetric reconstruction uses the plate rotation model (Matthews et al., 2016a) with a paleomagnetic reference frame (Van Hinsbergen et al., 2015a). We note that there can be a range of uncertainties regarding the absolute location of geographical constellation of continents. This particularly affects the latitudinal positions of the key gateways, depending on
the choice of plate tectonic model and reference frame. This can have significant consequences for the results of the ocean model simulations (e.g. (Baatsen et al., 2018)). However, as our paleobathymetry uses the same plate tectonic setup (model, reference frame) as the bathymetry used by the coupled climate model that provides the surface forcing Hutchinson et al. (2018), we expect our model simulations to at least be internally consistent.



### 4.3 The transport of the proto-ACC and its sensitivity to changing wind stress magnitude


The proto-ACC transport in our 1500_max_63°S case (TG transport: 47.9 Sv and DP transport: 38.3 Sv) is only about 30% of the modern ACC transport. Our experiments using a doubled wind stress (1500_max_63°S_dbw case) produce a more vigorous proto-ACC, with a TG

transport of ~75.4 Sv and a DP transport of ~44.8 Sv (Table 2). We find that the TG throughflow is strengthened significantly, but not doubled with doubled wind stress (the increase is ~57%). In contrast, the DP transport is relatively insensitive to an increase in wind strength, with the increase being only ~16%. This may be due to as eddy saturation (see Section 1.2), which has been found in many models of the Southern Ocean. Eddy saturation is a

consequence of mesoscale eddies in the Southern Ocean strengthening due to the increased momentum and energy input from the wind (Marshall et al., 2017). This enhanced eddy field is able to offset the expected steepening of isopycnals due to an increase in wind stress, making the ACC transport relatively insensitive to winds (Straub, 1993; Hallberg and Gnanadesikan, 2001; Tansley and Marshall, 2001). Munday et al. (2013) have proposed that the ocean model

with finer resolution will significantly reduce the sensitivity of circumpolar transport to wind stress compared to an eddy-permitting model. They even demonstrate zero sensitivity of their model's circumpolar transport to wind stress due to strong eddy activity in an eddy-resolving model. Our experiments show a less eddy-saturated result in the TG and DP transport, which may, in part, be due to our eddy-permitting model.


The changes in the sensitivity of current transport may be associated with the changes in some dominant terms of the zonal momentum budget. Munday et al. (2015) find that the addition of continental barriers into channel with a single ridge high enough to block f/H contour reintroduces the sensitivity of circumpolar transport at low winds, when their peak wind stress

is below 0.2 N m$^{-2}$. They analyze the zonal momentum budget and argue that the continental form stress may dominate the role of bottom form stress in the momentum balance with the introduction of continental obstacles. In our simulations, we can interpret sensitivity/insensitivity of the TG/DP transport via the momentum balance between wind stress and topographic form stress. The TG transport is dominated by subtropical gyres that sustain

the insensitive shallow TFS. This insensitive shallow TFS cannot adequately balance the doubled momentum input from wind stress, which acts to accelerate or strengthen the TG through-flow so that the TG transport is sensitive to the doubled wind stress. In contrast, the complicated submerged bathymetry in the DP region allows the DP through-flow to generate a deep TFS, which is expected to balance the wind stress and allow strong eddy saturation of

the DP throughflow, making DP transport insensitive to the doubled wind stress.

### 4.4 Zonal momentum balance and topographic form stress contributions

The zonal momentum balance is used to understand the role of wind stress and its variation in

the Late Eocene Southern Ocean. It was first proposed by Munk and Palmén (1951) that momentum input from the wind is balanced by pressure differences across bathymetric features (bottom form stress). Previous modelling studies confirm this balance with both idealized (Mcwilliams et al., 1978; Tréguier and Mcwilliams, 1990; Wolff et al., 1991; Marshall et al., 1993) and realistic bathymetry (Gille, 1997; Masich et al., 2015; Zhang and Nikurashin, 2020).

These have shown that the topographic form stress balances wind stress in the ocean model with idealized or realistic bathymetry. Our results show that this same balance prevails for Eocene bathymetry.



We further investigate how the depth integrated topographic form stress responds to the latitudinal migration and strength doubling of the wind stress. This response is separated into three different regions -northern, middle, and polar region. It is shown that the positive topographic form stress signal in the whole Southern Ocean above $H_s$ is mostly contributed by the oceanic gyres such as the subtropical gyre in the northern region. Below $H_s$, the pressure gradient across continents and submerged bathymetry both contribute to the negative topographic form stress required to cancel the positive topographic form stress signal sustained by the gyres. At $H_m$, these two contributions balance each other. As wind stress shifts southward, $H_m$ deepens (e.g. from 610 m in the 1500_max_53°S case to 2600 m in the 1500_max_63°S case). The southward migration of wind stress induces the subtropical gyre to expand in size. Accordingly, the maximum positive topographic form stress signal at $H_s$ also increases, potentially becoming larger than wind stress. Hence, the deepening of $H_m$ is required to balance this extra positive signal. The proto-ACC in the circumpolar belt flows across submerged bathymetric features like TG, DP, and the mid-ocean ridges. These features contribute to the increasing negative signals below $H_m$ for the whole Southern Ocean until the ocean bottom is reached. $H_d$ occurs where negative topographic form stress balances the wind stress. The deepening of $H_d$ in the 1500_max_63°S case may be associated with the development of the proto-ACC.

We use $H_s$ to decompose the total topographic form stress into shallow/deep components. In the northern part of the Late Eocene Southern Ocean, subtropical gyres generate a pressure gradient across continents to sustain the shallow TFS. This is the primary sink of the momentum input from subtropical easterly wind. The shallow TFS in the latitudes of subtropical easterlies is sensitive to the doubled wind stress. In the rest of the Southern Ocean, the proto-ACC dominates the ocean circulation and flows across submerged topography, with an associated pressure gradient. This pressure gradient contributes to the strong deep TFS to balance westerly wind stress. The deep TFS in the circumpolar belt is also sensitive to doubled wind stress. With the westerly wind peaks at 53°S, the positive signals of the shallow TFS in the ACC latitudes (around 60°S ~75°S) is contributed by the subpolar gyre. With the maximum westerly wind is at 63°S, the inception of the proto-ACC causes the shrink of the subpolar gyre, which removes the shallow positive TFS in the ACC latitudes.

## 5   Summary

Early studies proposed a connection between the opening/deepening of tectonic gateways linking the main basins of the Southern Ocean and changes in E-O oceanographic circulation patterns (Kennett, 1977; Murphy and Kennett, 1986; Toggweiler and Bjornsson, 2000; Huber et al., 2004; Sijp and England, 2004; Stickley et al., 2004). Recent studies further show the important role of the gateways' opening/deepening and wind stress in the changes of Southern Ocean gyres and the development of the ACC (Munday et al., 2015; Scher et al., 2015; Sauermilch et al., 2021). Here we use a high-resolution ocean model with a realistic paleo-bathymetry to investigate the sensitivity of the E-O Southern Ocean to TG deepening and changing wind stress. This study also analyzes the zonal momentum budget of the Southern Ocean to interpret the simulated dynamics. Some key findings are shown as following:

1. In the E-O Southern Ocean, southward shifts of wind stress expand the size of subtropical gyres and shrink the size of subpolar gyres.
2. When TG is shallow (300 m), the migration of wind stress does not cause major oceanographic changes. When TG is deep (1500 m), only the latitudinal alignment of



the maximum westerly wind with both TG and DP leads to a strong increase in proto-ACC transport and hence the thermal isolation of Antarctica.

3. The momentum input from doubled wind stress leads to a large increase of the proto-ACC transport through the TG, while the response of the transport through the DP is weak.

4. Subtropical gyres sustain the positive topographic form stress (the same direction with wind stress) at ocean surface (<700 m). The momentum balance between wind stress and topographic form stress occurs at depth (>3800 m) to support the strong proto-ACC in the circumpolar belt (50°S-70°S).

**Data availability**

The model data can be downloaded from Australian National Computational Infrastructure, an email can be sent to Qianjiang Xing (Qianjiang.xing@utas.edu.au) for access.

**Author contribution**

The model configuration was established by Andreas, Dave, and Jo, Isabel performed the spun-up simulation and Qianjiang changed the wind stress and performed further simulations. Qianjiang analyzed the results and finished the manuscript with supervision by Andreas, Dave, and Jo. Isabel helped to plot figure 1 and gave a lot of suggestions on the manuscript. All authors reviewed this manuscript.

**Competing interests**

The contact author has declared that neither they nor their co-authors have any competing interests.

**Acknowledgements**

This work was funded by the Australian Research Council Discovery Project (DP180102280). We acknowledge the Australian National Computational Infrastructure Merit Allocation Scheme projects used for running the model simulations. We are grateful for discussions with Dr. Edward Doddridge on ocean modelling and Xihan Zhang on estimating topographic form stress. IS is supported by the ARC Discovery Project (DP180102280) and ERC starting grant OceaNice (#802835).



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
