# Peer review of "The sensitivity of the Eocene-Oligocene Southern Ocean to strength and position of wind stress"

_Climate of the Past, 2022_

## Author Comment (AC2)

Response to referee 2

The manuscript addresses the question of the Eocene-Oligocene Southern Ocean and its sensitivity to wind stress strengthening and widening/deepening of the Tasmanian Gateway and Drake Passage in setting up an ACC. The paper is very interesting and provides a solid demonstration of the momentum balance at play through an analysis of the zonal momentum balance and its different terms. The results, subject to all possible limitations and caveats, are convincing. However, I found the paper poorly written, very long and repetitive at times. I have the impression the same message and results can be conveyed with perhaps half of the text, improved figures and a more structured discussion/summary.

***We are grateful to the referee for their positive and helpful comments. We will address the proposed problems and strengthen our paper by improving its clarity and conciseness. Our response is given below in bold and italic.***

Please find below a list of suggestions, questions and corrections.

L56 New and improved estimates could be used here: Koenig et al. (2014) estimated a full depth transport of 141 ± 2.7 Sv and Chidichimo et al. (2014) and Donohue et al. (2016) estimated a full depth transport of 173.3 ± 10.7 Sv.

***Response: Thanks for suggesting these recent estimations. We will modify this sentence as "...the strongest ocean current, with a full depth transport of 137 ± 7 Sv (1 Sv=106 m3s-1) at DP (Meredith et al et al., 2011). It is further estimated with a full depth baroclinic transport of 127.7 ± 1.0 Sv (Chidichimo et al., 2014) and full depth total transport of 173.3 ± 10.7 Sv (Donohue et al., 2016).***

L115 This is something that you could easily check and should be shown to test the regime change from subtropical gyre dominated to a proto-ACC: please add an analysis of the ocean heat trasport and its eddy contribution.

***Response: We will add our analysis of meridional heat transport in the revised manuscript to show the sensitivity heat transport for mid-high latitudes due to the inception of proto-ACC. This will increase the length of our paper. However, we will work to ensure that this does not harm its clarity.***

L154-155 Please rephrase, something is odd here.

***Response: We will delete "remains unknown".***

L240 Why do you use a model with no sea-ice? In understand and appreciate the idealized framework of a regional configuration but I don't see what is gained here by eliminating all

possible feedbacks induced by sea-ice. Also, the model of Hutchinson et al, 2018 presumably uses sea-ice (CM2.1), so your surface restoring has that infomration.

*Response: Around 38 Ma, the observed global SST was in the range of 25 to 19.5 °C (Bijl et al., 2009; Liu et al.,2009; Houben et al., 2019). The minimum SST of our simulations is around 11°C, which reflects the warm zonal-annual-mean conditions in Hutchinson et al.'s model. These warm surface temperatures remove the possibility of sea-ice in our model, and so we have not included a sea-ice component.*

*(Bijl, P.K., Schouten, S., Sluijs, A., Reichart, G.J., Zachos, J.C. and Brinkhuis, H., 2009. Early Palaeogene temperature evolution of the southwest Pacific Ocean. Nature, 461(7265), pp.776-779.*

*Liu, Z., Pagani, M., Zinniker, D., DeConto, R., Huber, M., Brinkhuis, H., Shah, S.R., Leckie, R.M. and Pearson, A., 2009. Global cooling during the Eocene-Oligocene climate transition. science, 323(5918), pp.1187-1190.*

*Houben, A.J., Bijl, P.K., Sluijs, A., Schouten, S. and Brinkhuis, H., 2019. Late Eocene Southern Ocean cooling and invigoration of circulation preconditioned Antarctica for full-scale glaciation. Geochemistry, Geophysics, Geosystems, 20(5), pp.2214-2234.)*

L248 You use a relatively strong SST and SSS restoring of 10 days. How is that affecting your simulations and results when you try to initiate a thermal isolation of the Antarctic?

*Response: Due to the strong feedback between SST and surface heat flux, 10 days is a fairly standard restoring time for SST in many applications. In contrast, for salinity there is only a weak feedback between SSS and evaporation/precipitation. As a result, this is quite a short timescale for salinity restoring. Our aim in using 10 days for both temperature and salinity was to ensure a good fit to Hutchinson et al.'s SST and SSS fields.*

*In general, a short restoring time scale in the model tends to lead to higher fluxes and acts to damp surface variance in temperature/salinity, as well as eddy kinetic energy. As shown by Zhai & Munday (2014), this also leads to a larger sensitivity to wind stress of the overturning, which is one reason why we have not examined the MOC in this paper. Zhai & Munday do not comment on the sensitivity of the circumpolar transport. However, examination of their figures indicates surface restoring makes the isopycnal slopes less sensitive to wind stress, which implies that the thermal wind transport is as well. In our case, the thermal wind transport is the majority of the zonal flow through TG and DP. Therefore, we can expect that the sensitivity we report is lower than might be achieved in a model that used a pure flux condition on temperature and salinity.*

*In our revision we will add a short discussion of the surface restoring to the Methods section, as a response to both reviewers. This will include a short version of the above discussion.*

*(Zhai, X. and D. R. Munday, (2014). Sensitivity of Southern Ocean overturning to wind stress changes: Role of surface restoring time scales, Ocean Modelling, 84, 12-25).*

L281 I am not sure about a regional configuration, but a spin-up of 80 years and sensitivity experiments of 60 years seem a little short to me. It would be intersting to see time series of different metrics to show the circulation is stable and how it changes with the deepening of gateways and shifting of winds.

*Response: The model was spun-up for 85 years by Sauermilch et al. (2021). After changing to our revised wind stress, we ran our experiments for an additional 60 years to adjust to the wind stress conditions. In terms of circumpolar transport, our simulations are well equilibrated. We will add the following figure of a time series of zonal transports to the Appendix of the revised paper. This shows a good degree of equilibration, according to this metric, for most of our experiments.*

L284 Another point related to the model configuration: I am not sure what the actual shape of the zonal wind stress is. Is it a zonal mean and you simply shift it north and south? It is not clear form the text whether zonal wind stress is zonally dependent. Presumably that would matter in terms of alignments with the gateways and relative strength at the DP and TG.

*Response: We apologise for the lack of clarity on the form of the wind stress. It is, indeed, a zonally-symmetric wind stress. It has been smoothed slightly, relative to Sauermilch et al. (2021), in order to make adjusting it to the north and south cleaner. We will make this point clearer during revision.*

*A non-zonally-symmetric wind stress would indeed raise interesting questions regarding the alignment with TG and DP. We will raise this point in our revised discussion, thank you for suggesting this idea.*

L360 details of the discretization, also in L634, should go into the supplmentary information (Eq. 4 is already present). Also, Eq. S3 is missing the $1/\rho_0$.

*Response: Thanks for spotting the missing $1/\rho_0$. We will add more detail of the algorithm to the Appendices of the revised paper. The revised paper will read, at this point; "Following Masich et al. (2015), the zonally vertically integrated total zonal pressure gradient (or total topographic form stress) is discretised as per Eq (4). We extract the total topographic form stress from the zonally vertically integrated total zonal pressure gradient field. More detail on the calculation of topographic form*

*stress, and errors associated with the use of partial model cells, can be found in the Appendix section 1 and 2 and method section of Masich et al. (2015)."*

L415 Fig.4 is really difficult to read with its present choice of coulours and arrows and should be improved. Consider a specific countour for the SST to highlight the change in temperature along the coast, and different/fewer arrows. Also perhaps less panels

*Response: We will remove the arrows and add the suggested contour to improve the clarity of this figure.*

L478 Eg. 5 is missing

*Response: Thanks for spotting the missing equation, we will adjust the numbers appropriately.*

Figure 7 This figure is also difficult to read. Consider adding to the same panel both the normal and doubled wind stress to highlight differences.

*Response: Thank you for the idea on how to modify this figure. We will combine the panels as suggested and alter the number of lines so as to improve clarity.*

L632 Eq. 5 is missing as well as section 2.5

*Response: Thank you for spotting these errors. We will adjust the numbers appropriately.*

L665 I really like your results but the Discussion section is difficult to read, repetitive and often not a 'Discussion' but rather a 'Summary'. Please improve your text to ease the read.

*Response: Thank you for your positive opinion. We will work to improve the structure and flow of the Discussion. We will remove any repeated material and seek to improve its clarity.*

---

## Author Response (AR1)

**Statements about changes in the revised manuscript**

Based on useful comments from two reviewers, we have made many changes to our manuscript. We used Word to write the first draft of the manuscript and changed to LaTeX to write the revised manuscript to obtain better formatting. Therefore, it is hard to use track changes in Word or latexdiff in LaTeX to show what we have changed. However, we have manually highlighted the changes in the revised manuscript and appendices. The detailed changes relevant to each reviewer's comments have been listed in the context of response to referees. The main changes of the revised manuscript are summarised below:

1. Changed "thermal isolation of Antarctic " to "cooling of Eocene Southern Ocean". See Page 1, Line 31 and Page 17, Line 40 in the revised manuscript. This responds to the fourth main comment of reviewer 1.
2. Changed titles of subsections 1.1,1.2, and 1.3 in Introduction. See Page 1, Line 51, Page 2, Line 54, and Page 3, Line 1. This responds to the second main comment of reviewer 1.
3. Reduced the text on bottom form stress/topographic form stress in the Introduction and Method. Moved them into Results (Section 3.3) and Appendices. This responds to the second main comment of reviewer 1.
4. Explain clearly in Section 2.3 (Page 5, Line 11-24) why we only shift the wind stress to represent the relative position between gateways and wind stress. This responds to the first and fifth main comments of reviewer 1.
5. Added a figure of meridional heat transport (MHT) and some relevant text/analysis in Section 3.2. This responds to comments from both reviewers.
6. Changed titles of subsections 4.1, 4.2, and 4.3 in Discussion. See Page 15, Line 9, Page 15, Line 105, and Page 16, Line 49. This responds to the last comment from reviewer 2.
7. Removed repetitive text from Discussion. This responds to the last comment from reviewer 2.
8. Added a subsection 4.1 of Discussion, discussing some uncertainties of the model configuration. This responds to the last comment from reviewer 2.
9. Refresh the text explaining why TG and DP transport show different responses to doubled wind stress in Section 4.3.
10. Modified most figures according to both reviewer's comments.
11. Added some context surrounding zonal momentum balance, a figure of reconstructed paleo-bathymetry, and a figure of time series of TG transport in the Appendices.

**Response to Referee 1**

The authors present the results of eddy-permitting ocean-only simulations to shed light on the development of a proto-ACC around the Eocene-Oligocene Transition. In contrast to most earlier work, they provide a substantial increase in model resolution and a more realistic set of model configurations to represent the effects of a gradual gateway opening. A very detailed analysis is made of the momentum balance and the overall results are of great value to understand the role of Southern Ocean Gateways in much of the Cenozoic climate. Regardless, the manuscript still needs work to clearly present/explain the scientific set-up of the experiments, their motivation, the results and their implications.

The manuscript is generally well written and mostly free of errors, especially the figures should be adjusted to improve readability and clarity.

***We thank Michiel Baatsen for his many constructive comments and for highlighting the important contributions of our paper. We have carefully considered all his comments and suggestions to improve the manuscript. Our response is given below in bold and italic.***

**Main comments**:
- I am missing additional background, motivation and some explanation of the choices for wind stress profiles/latitudes (other than referencing to Scher et al 2015).

  ***Response: As shown in Figure 1 of our paper, the Tasman Gateway (TG) rapidly widened and its paleolatitude moved northward from ~58°S (38 Ma) to ~53°S (28 Ma). Meanwhile, the northward movement of Drake Passage (DP) is relatively slow, as its paleolatitude only moved northward by 1-2 degrees from 38 Ma (about 63°S) to 28 Ma (about 61°S). The northward movement of the gateways relative to the wind stress can be expected to impact the Eocene Southern Ocean and proto-ACC. However, changing the paleobathymetric reconstruction in our model is a costly and time-consuming process. In contrast, adjusting the wind stress' paleolatitudes is simple and much quicker, allowing us to test the hypothesis that the alignment of the wind with the gateways is a key part of the proto-ACC's development.***

  ***In the classical theory of wind-driven gyres, wind stress and continental barriers sustain the large-scale gyres in the oceanic basins (Munk, 1950). The latitudinal position of gyres boundaries is aligned with the position of zero wind stress gradient, which is also typically the position of maximum wind stress (Sverdrup, 1947). The southward movements of wind stress can narrow the spatial scale of subpolar gyres under the restriction of the Antarctic continent, while northward wind shifts extend the subpolar gyres' spatial scale. Focusing on the Southern Ocean, the shifted wind stress could influence the position of gyres boundaries and the pathway of the proto-ACC. Some studies have indicated that the latitude of maximum wind stress can have a large influence on the position of the ACC. For example, Allison et al. (2010) used an idealized model to show that the circumpolar current adjusts its position to align with the shifted wind jet, except when the current is forced to turn poleward to penetrate Drake Passage.***

  ***Motivated by this, our study uses a smoothed version of the zonal average wind stress from Sauermilch et al. 2021 to highlight the position of the peak wind stress. From Figure 1, we know that the TG had ~5° northward movement during the E-O transition. As such, we select a 5° shift as the perturbation for our wind stress. In addition, we conduct experiments with a***

*southward 5° shift and a southward 10° shift as further perturbation experiments. These experiments are intended to simulate the changes of relative positions between wind stress and ocean gateway.*

*The applied bathymetry in this study is reconstructed to 38 Ma, the latitudinal positions of Southern Ocean gateways (TG and DP) are shown in Figure 1. We do not change the latitudinal position of TG and DP in this study, but we manually adjust the gateways depths (TG and DP) in the paleobathymetry grids, with the depth values referring to the shallowest part of each gateway. We chose 300 m and 1500 m for TG depth, 1000 m for DP depth to simulate the impact of TG deepening.*

*We have ensured that this background and motivation is clearly highlighted in our revisions. To do so we have revised the Introduction and the Methods sections (see Page 3, Line 102 - Page 5, Line 5; Page 5, Line 11-24). We aimed to remove any ambiguity over the changing bathymetry vs. moving wind stress.*

*(Allison, L.C., Johnson, H.L., Marshall, D.P. and Munday, D.R., 2010. Where do winds drive the Antarctic Circumpolar Current?. Geophysical Research Letters, 37(12).)*

- It is not always clear which questions are asked, what the hypotheses are and how they are answered.

There is an extensive introduction and study of the different components in the momentum balance, but it is tough to see the role these play in the larger picture. Especially the part on bottom form stresses is rather tedious to read and does not seem to answer many questions. In general, the manuscript is quite lengthy and lacking some clear structure/connections to see the overall story. It may therefore be better to focus on some specific topics, rather than treating all aspects in such detail.

*Response: The introduction aims to introduce two drivers that would influence the onset and strengthening of the proto-ACC; Southern Ocean gateways deepening and shifts in wind stress position/strength. Our paper is then aimed at addressing whether shifted wind stress in position/strength, in the context of ocean gateway opening, have an impact on the early Cenozoic Southern Ocean and proto-ACC.*

*Understanding the zonal momentum balance is an important step in understanding the dynamics of the proto-ACC. In the zonal momentum balance of the modern Southern Ocean, bottom form stress is known to be the primary sink for momentum input via zonal wind stress. However, bottom form stress has not been widely considered in palaeoceanography, where changing continental configurations and bathymetry may impact the prevailing momentum balance.*

*We aim to consider the balance between wind stress and bottom form stress in the late Eocene Southern Ocean and how this balance is associated with the proto-ACC transport. As such, we feel that a thorough introduction to this momentum balance is required in the paper. However, we recognise that this may be too much detail in the introduction to the paper as a whole. As such, we have altered the layout and moved much of this material into the main body of the paper itself (Page 11, Line 14 - page 12, Line 25) and Appendix. We also provide additional subheadings and additional linking*

*sentences between different subsections in the revised manuscript. In revising the Introduction, we have ensured that our research questions are clear (Page 2, Line 101-106 and Page 3, Line 27-29) and use our Discussion/Conclusions section to provide clear answers (Page 16, Line 42-44 and Page 17, Line 2-9).*

- Using an ocean-only model is a big limitation, especially regarding the feedback between temperatures and wind stress as well as missing the atmospheric component of meridional heat transports. As they are restored to fixed distributions (of which the treatment and implementation could use some more explaining), sea surface temperatre and salinity fields are challenging to interpret and one should be careful drawing conclusions from these.

  *Response: We agree that the use of an ocean-only model is indeed a limitation and have added a discussion of this in our revision at subsection 4.1 of Discussion.*

  *Coupled atmosphere-ocean models usually have low resolution ocean components. This limits their representation of ocean circulation/dynamics and also impacts their ability to accurately represent the detail of the sea floor. These are all essential ingredients in modern Southern Ocean dynamics. We have chosen to restrict ourselves to an ocean-only model in order to use a higher resolution model that can more accurately model the Southern Ocean. This compromise does remove key feedbacks, which we have discussed at Page 15, Line numbers 41-54 in our revision. The Supplementary Information of Sauermilch et al. (2021) shows that changing resolution or bathymetry has a large impact on the model results, which we have highlighted in the Discussion of our revised manuscript (see Page 15, Line 17-21).*

  *We also include more information regarding the surface restoring conditions, which is implemented as a form of Haney (1971) relaxation to a surface air temperature (see Page 15, Line 41-69).*

  *(Haney, R. L., 1971: Surface thermal boundary condition for ocean circulation models. J. Phys. Oceanogr., 1, 241–248.)*

- Decreasing SSTs in the Southern Ocean are said to be an indicator of the thermal isolation of Antarctica.

  I am missing a clear explanation how those SSTs would be representative of temperatures on the Antarctic continent. Apart from a single figure showing SSTs, I am missing an assessment of meridional heat transports and how these would be linked to Antarctic temperatures altogether. What happens in the ocean is no doubt interesting and relevant, but statements regarding Antarctic temperatures as a whole are not well supported by the results presented here.

  *Response: We agree that SST is not a reliable indicator of Antarctic temperature and have amended such statements to "cooling of the Eocene Southern Ocean" (See Page 1, Line 31 and Page 17, Line 40). We also add a figure (Figure 4) and analysis (Page 6, Line 23-47) of the model's meridional heat transport in the revised manuscript. This indicates substantial changes in heat transport due to wind stress shifts and changing TG depth. We can find the convergence of heat south of 50◦S in the 300_max_63◦S case is typically higher than the other cases. This indicates that the ocean transports more heat into this region and sustains higher temperatures. Corresponding*

*to this, the domain average sea surface of the 300_max_63∘S case is warmer than other three cases. In contrast, the 1500_max_63∘S case has the weakest southward MHT, thus the minimal heat convergence south of 50∘S. This is due to the strongest circumpolar current blocking gyre driven heat advection.*

- I am missing a section on the geographical configurations used, as shown in Figure 1, motivating the different time intervals and explaining the different Gateway configurations. This makes it hard to interpret many of the results.

  *Response: In our experiments, we have only applied 38 Ma bathymetry and we apologise for any confusion arising from Figure 1. Figure 1 shows the evolution of paleo continents in different periods (38 Ma, 30 Ma, 28 Ma), which is intended to help readers understand that Southern Ocean gateways were undergoing northward movement or widening during late Eocene to early Oligocene (see Page 3, Line 98-102). So the moving oceanic gateways probably aligned with the peak wind stress to influence the Southern Ocean current pattern. As mentioned in the above response, simulating different paleobathymetric conditions is difficult and expensive. As we seek to test the impact of relative positions between gateways and peak wind stress conditions, we restrict ourselves to one paleobathymetry whilst applying different wind stress conditions (see Page 5, Line 17-24).*

**Specific comments**:
- L53: maybe trivial, but good to specify that this is for the present configuration

  *Response: We have modified this to read "Page 1, Line 36: The present Southern Ocean …"*

- L57: are there any more recent observational-based estimates of ACC strength?

  *Response: We have modified this sentence to read "Page 1, Line 40: The ACC has a volume transport of 127.7 ± 1 Sv (1 Sv=106 m3s−1, Chidichimo et al., 2014) to 137 ± 7 Sv at Drake Passage (DP) (Meredith et al., 2011), 141 ± 13 Sv from in situ and satellite observations (Koenig et al., 2014) or 173.3 ± 10.7 Sv when the near bottom flow is included (Donohue et al., 2016)."*

  *(Chidichimo, M.P., Donohue, K.A., Watts, D.R. and Tracey, K.L., 2014. Baroclinic transport time series of the Antarctic Circumpolar Current measured in Drake Passage. Journal of Physical Oceanography, 44(7), pp.1829-1853.)*

  *(Donohue, K.A., Tracey, K.L., Watts, D.R., Chidichimo, M.P. and Chereskin, T.K., 2016. Mean antarctic circumpolar current transport measured in drake passage. Geophysical Research Letters, 43(22), pp.11-760.)*

- L60: I see you refer to figure 1 here, but this is mainly showing the applied wind stress forcing. I don't mind this reference, but it seems more intuitive to move this figure further down.

  *Response: We have moved this figure to the Methods section (Page 4) of the revised manuscript to aid clarity.*

- L100: This was already greatly improved in more recent model efforts, see e.g. Hutchinson et al. 2018, Kennedy-Asser et al. 2020, Baatsen et al. 2020. The latter shows a comparison to the Huber et al 2004 results, with subtropical waters reaching much of East Antarctica in the newer CESM simulations.

*Response: Thanks for highlighting these papers, we have added these additional references in the revised manuscript: "Page 2, Line 30: Thus, Huber et al. (2004) propose that insufficient warm water from the subtropics reaches high latitudes to keep Antarctica warm prior to E-O transition. However, this hypothesis is at odds with some recent modelling efforts (Hutchinson et al. 2018, Baatsen et al. 2020, Sauermilch et al., 2021)"*

- L116: This is a very nice paper, you could also consider referring to Viebahn et al. (2016) who did a similar experiment (although using PD bathymetry) comparing HR/LR simulations.

  *Response: Thanks for the suggestion. We have included this reference in the revised manuscript; "Page 2, Line 47: These new results contrast with Huber et al. (2004)'s results by showing that substantial heat transport from the subtropics to Antarctica is enabled by the subpolar gyres (Sauermilch et al., 2021), which is also consistent with the subpolar gyres-driven warming around Antarctica in the model study of Viebahn et al. (2016) although using present-day bathymetry."*

- L129: Baatsen et al. 2020 already find a ~45Sv Tasmanian Gateway Transport with relatively shallow (500-1000) TG and DP, see supp. Figure 3.

  *Response: Thanks for the suggestion. We have included this reference in the revised manuscript; "Page 2, Line 74: Hill et al. (2013) simulates a 44 Sv volume transport across DP at 32 Ma and a strong proto-ACC (transport of >90 Sv) is established after 26 Ma, although Baatsen et al. (2020) simulate a 45Sv TG throughflow transport with 38 Ma geography reconstruction and shallow TG. Hence, the tectonically…"*

- L145: I would expect westerly winds to be aligned with the polar front, do the authors refer to the oceanic (i.e. SST) front? Does this infer a mismatch between atmospheric/oceanic polar fronts?

  *Response: The polar front in the Scher et al. (2015) refers to the boundary between the polar easterly and mid-latitude westerly winds during the Oligocene, rather than the oceanic polar front. Scher et al. (2015) intended to test the alignment between the northern margin of the Tasmanian Gateway and north of the Oligocene atmospheric polar front. This may well cause a mismatch between atmosphere and ocean; we have clarified this in the revised manuscript; "Page 2, Line 92: Scher et al. (2015) propose that the relative latitudinal position of the westerly winds and Southern Ocean gateways is another key factor in proto-ACC development. They compare the relative location of the Oligocene TG to the position of the polar front (the boundary between polar easterlies and mid-latitude westerlies) and suggest that the delayed onset of ACC-like flow is due to their misalignment."*

- L162: It is unclear to me from the text what exactly is meant by eddy-saturation and what it results to.

  *Response: According to Marshall et al., (2017), eddy saturation is the phenomenon where the ACC volume transport is relatively insensitive to changing surface wind forcing in high resolution models. This is due to the (partial) resolution of ocean eddies, leading to increased power input increasing the eddy energy, instead of the mean flow. Our model is eddy-permitting and we should consider the occurrence of eddy saturation in our*

***estimated transport of the proto-ACC. We have clarified this section of the introduction during revision of the paper (Page 3 Line 3-6).***

***(Marshall, D.P., Ambaum, M.H., Maddison, J.R., Munday, D.R. and Novak, L., 2017. Eddy saturation and frictional control of the Antarctic Circumpolar Current. Geophysical research letters, 44(1), pp.286-292.)***

- L174: Does 0.2N/m^2 agree with observed wind stress across the Southern Ocean? If so, is there an explanation for the underestimation of ACC strength in these idealised simulations?

  ***Response: Yes, 0.2N/m^2 agrees with observed peak wind stress across the Southern Ocean according to Lin et al. (2020). Although the wind stress of the Southern Ocean is notoriously hard to observe. The underestimation of ACC strength in this case may be due to a variety of reasons related to the model domain. For example, Munday et al. (2015) use a channel model with a depth of 3000m and enhanced diapycnal diffusivity near the northern boundary. The first of these reduces the cross-sectional area of the model, reducing the transport due to near-bottom flow. The second influences the stratification throughout the domain, and so directly influences the transport.***

  ***(Lin, X., Zhai, X., Wang, Z., & Munday, D. R. (2020). Southern Ocean Wind Stress in CMIP5 Models: Role of Wind Fluctuations, Journal of Climate, 33(4), 1209-1226)***

- L200: I expect this also depends on the depth of the bathymetry ridge?

  ***Response: Yes, different ridge depths can influence the bottom form stress, causing the insensitivity of zonal volume transport, which is also a part of our study. In addition, ocean currents are strongly constrained by f/H contours. The submerged topography can block f/H contour, which steers the current to maintain conservation of potential vorticity. The blocking of f/H contours reduces the velocity below the bathymetric level and allows the transport due to thermal wind shear to dominate the ocean current transport. Different ridge depths can decide the dominance of thermal wind transport, which also influences the sensitivity of zonal volume transport.***

- L225: Is 0.25deg resolution sufficient to be eddy-permitting across the region of interest? For this, I would like to see a comparison of e.g. the local Rossby radius of deformation and model resolution.

  ***Response: There is no precise definition of eddy-permitting vs. eddy resolving, in part due to different models representing models in different ways. Typically, the literature considers 1/4° to be eddy-permitting and 1/10°-1/12° to be eddy resolving. The key difference being that, for the modern ocean, 1/4° has about half the eddy kinetic energy of the ocean as measured by satellite altimetry and 1/12° reaches parity, see, for example, Delworth et al. (2012). The eddy activity of our model can be seen in our figures of streamfunction and temperature, indicating that there is a reasonably strong eddy field.***

  ***Given concerns regarding the length and conciseness of the paper, as raised by reviewer 2, we would prefer not to include this suggested figure in the paper. However, we include Figure 1 of Hallberg (2013) below. This shows the model grid spacing required to resolve the deformation radius with two***

*grid boxes, i.e. what could be considered eddy resolving. This shows that at high southern latitudes, 1/4° is too coarse to fully resolve the eddy field, and so is best thought of as eddy-permitting.*

[Figure]

See e.g. LaCasce and Groekamp 2020, who show a first surface mode deformation scale of 10-20km at 60S in the present-day ocean.

*At our model resolution, the grid spacing at 60°S is ~12.5km. Given that eddies are typically several multiples of the deformation radius, when fully mature, we believe this puts our model firmly in the eddy-permitting category. If anything, we would expect weaker stratification, and therefore larger deformation radii, due to reduced temperature gradients in the Eocene ocean. This would imply a slightly better resolution of the eddy field than in the modern ocean.*

*Most of these contexts have been included into the Discussion of revised manuscript (Page 15, Line 25-40).*

*(Delworth, T. L. et al. (2012). Simulated cliamte and climate change in the GFDL CM2.5 high-resolution coupled climate model, Journal of Cliamte, 25, 2755-2781.)*

*(Hallberg, R., (2013). Using a resolution function to regulate parameterizations of oceanic mesoscale eddy effects, Ocean Modelling, 72, 93-103.)*

- L238: a visual representation of the model domain, resolution and boundary conditions would be very helpful here.

*Response: Figure A3 in the appendix shows the model domain and bathymetry. We have added two more panels showing the bathymetry of a 300m and 1500m TG, whilst ensuring this figure is properly referred to earlier in the paper. This figure is now in the Appendix D, Figure D1 (Page 23) of the revised manuscript.*

- L265: Does this mean the authors use a continental slope from paleogeographic reconstructions, or present-day observations? Especially the East-Antarctic margin (both extent and slope) has changed significantly since the Eocene

  *Response: Yes, we use paleogeographic reconstructions. There is a more detailed description on the calculation of continental slope reconstruction in Hochmuth et al. 2020. We apply McKenzie (1978)'s subsidence model for the extended continental crust.*

  *(McKenzie, Dan. "Some remarks on the development of sedimentary basins." Earth and Planetary science letters 40.1 (1978): 25-32)*

- L288: I believe this is a very nice overview of possibilities, but I am struggling to find the motivation for these choices. Are there any simulations or theoretical considerations that motivate the applied shifts in max wind stress latitude?

  *Response: We have given a related response to the first main comments (see Page 5, Line 17-24).*

- L295: I am missing some information here on how well equilibrated the simulations are after 45 years, especially regarding the zonal transports and isopycnal slopes through TG/DP

  *Response: The model was spun-up for 85 years by Sauermilch et al. (2021). After changing to our revised wind stress, we ran our experiments for an additional 60 years to adjust to the wind stress conditions (see Page, Line 7-11). In terms of circumpolar transport, our simulations are well equilibrated. We have added the following figure of a time series of zonal transports to the Appendix E of the revised manuscript (see Page 24). This shows a good degree of equilibration, according to this metric, for most of our experiments.*

[Figure]

- L329: Does this mean v=0 is applied at the northern boundary of the domain? This would imply the complete absence of a meridional overturning cell extending beyond the model domain.

*Response: Yes, we apply zero meridional velocities at the northern boundary to maintain volume conservation in our model domain. The restoring condition at the northern boundary allows water to change temperature/salinity and upwell or downwell. This allows the model to represent water mass conversion taking place to the north of the model domain, also representing the meridional overturning. This prevents the model from collapsing to a condition of zero residual mean overturning. This technique has been applied in the modern case and does a good job of allowing small model domains to reproduce the local overturning of the Southern Ocean (Abernathey et al., 2011).*

*(Abernathey, R., J. Marshall, and D. Ferreira, 2011. The dependence of Southern Ocean overturning on wind stress, Journal of Physical Oceanography, 41, 2261-2278/)*

- L379: Also here, there are quite a few more recent simulations showing this

*Response: We agree and have corrected this omission with additional references in the revised manuscript; "Page 5, Line 51: ...anti-clockwise for the subtropical gyres, and clockwise for the subpolar gyres (Huber et al., 2004; Huber and Nof, 2006; Hill et al., 2013, Hutchinson et al. 2018, Baatsen et al. 2020, Sauermilch et al. 2021)"*

- L409: 'higher/lower' is a bit ambiguous, especially on the Southern Hemisphere, consider using 'equatorward/poleward' instead.

*Response: Thank you for highlighting this ambiguity., We have resolved this in the suggested way, e.g., "Page 5, Line 77:…where the northernmost latitude of the subpolar gyres has been restricted to latitudes poleward of 63°S.".*

- L460: At 300m, the relative strengthening of TG transport is larger with the more southerly wind stress max. This is reversed and much less sensitive to wind stress latitude at 1500m, can you explain why?

*Response: At 300m cases, the TG transport is small and its change due to doubled wind stress is also small in an absolute sense. As a result, these percentages may be misleading, which we have highlighted in the revised manuscript (Page 6, Line 66).*

*However, we can still give a broad explanation. From Figure 3, we can see that when the TG is at 300m, the 53°S peak wind stress condition allows both counterclockwise and clockwise gyres to cross TG. The doubled wind stress increases the strength of both gyres. However, because the net transport is a small residual of westward and eastward flow in a gyre, the absolute change in TG transport is small. The 63°S peak wind stress condition leads to a large weakening of the clockwise gyre driving net transport through TG. The doubled wind stress is able to reverse this weakening to a larger degree and so causes a larger relative increase in net transport through TG. In the 1500m TG case, the clockwise gyre for both 53°S and 63°S peak wind stress cases is weakened and the ocean is entering a regime that is more like the modern Southern Ocean. The net transport through TG is now a result of a true circumpolar flow, instead of the net transport of two strongly compensated flows of a gyre. As a result, the flow responds to changes in*

*wind stress in a more eddy saturated regime and the transport becomes less sensitive.*

- L484: Is there also a formal definition of what you refer to as 'nearly homogeneous'? Why does this separate the thermal wind component from the bottom slope contribution? A non-uniform bottom slope would imply zonal flow throughout the column.

  *Response: There isn't really a formal definition of 'nearly homogenous'. Here we mean that the gradients of density (temperature/salinity) should be weak. Since the thermal wind component of the zonal flow requires a strong meridional gradient of density, this allows us to cleanly separate the flow into bottom flow and thermal wind flow. The bottom flow is, broadly speaking, that due to the barotropic response to forcing, although this is not a strict barotropic/baroclinic decomposition. We do interpret the bottom flow as taking place throughout the water column, but it is not due to the bottom slope itself.*

  *We have clarified this in the revised manuscript, to ensure that the decomposition and its application is clear. For example; "Page 9, Line 4: we select a model level below which the current velocities have little vertical and meridional gradient (shear). This allows as clean separation into the flow that is due to thermal wind shear and that is not."*

- L508: The DP transport is substantially weaker than the TG one (this is also the case in Baatsen et al. 2020, using a similar paleobathymetry). What does this imply for a possible proto-ACC?

  *Response: It is not unreasonable to expect that a deeper gateway would allow for a stronger transport. In part this would be due to the potential for bottom flow to take place over more of the water column. In this case, some of the transport through TG is flowing northwards along the coast of Australia and recirculating as part of the subtropical gyre. This prevents it from having to flow through DP and may allow for a stronger connection between TG waters and the East Pacific than in the modern Southern Ocean.*

- L527: I would re-phrase this sentence and not refer to specific colours shown in a Figure quite a ways back at this point.

  *Response: Thank you for highlighting this issue. We have modified this sentence in the revised manuscript; "Page 19, Line 54: the total topographic form stress and total pressure gradient are almost equal as the of blue and red curves in the Figure B2 are almost coincident". Now this context is in the Appendix.*

- L540: Could you also suggest how?

  *Response: Masich et al., (2015) divide the total topographic form stress (TFS) for the whole basin into two parts, using 3700m as the boundary between these regions. From the surface to 3700m, the shallower TFS signals are mostly provided by continents and bathymetric features within ACC latitudes. Such shallower TFS signals can balance the surface wind stress. From 3700m to the seafloor, the lower TFS signals are generated by large-scale rough seafloor so that the lower TFS signals spread over the basin, with positive signals in some regions and negative signals in the rest. If conducting a zonal integral of the lower TFS, the deeper positive and*

*negative contributions tend to cancel. As a result the lower TFS signals balance themselves zonally, leaving the shallower TFS to balance the wind stress.*

- L665-689: This part seems fit better in the introduction rather than discussion?

  ***Response: We agree, and we have removed this part due to its repetitive context in the revised manuscript.***

- L707: If I understand correctly, the bathymetry used here changes both depth and latitude of the gateways at the same time.

  It may therefore be also the latitude offset between TG vs DP that may be just as important as their depth. I am missing this in the discussion.

  ***Response: As explained above, our experiments only change the wind stress position and the depth of TG. This is due to the difficulty and complexity of producing different paleobathymetric reconstructions to simulate the northward movement of ocean gateways. We apologise for this confusion and ensure that we have resolved the ambiguity in the revised manuscript, as laid out above (see Page 3, Line 102 - Page 5, Line 5).***

  ***We agree that the latitudinal offset of TG and DP could indeed play an important role. In the parlance of Munday et al. (2015), this could be due to a shift in the momentum balance, with bottom form stress and continental form stress influencing the sensitivity of the transport. This is an unfortunate omission that we thank the reviewer for highlighting. In the revised manuscript, we have added this point to the discussion (Page 17, Line 10).***

- L750: There are some very important nuances and limitations listed here, which would deserve more attention in the discussion but also up front.

  ***Response: We agree and as part of our revision have moved these important points to the first subsection of the discussion (Page 15, Line 84).***

- L780: you already explain the concept of eddy saturation in the introduction. Most of the remaining part of this paragraph is a repetition of what is said earlier.

  ***Response: As part of our revision, we have removed any repeated material and aim to make this section more concise.***

**Figures**:
- Fig1: the contrast between title and axes font size is quite large, it would seem right to adjust those somewhat.

  Although rather straightforward, the axes are missing labels/units as well. The figure now has a rather extensive caption, this could be shortened considerably by putting some of this information into legends/labels

  ***Response: Thank you for highlighting this. We have adjusted the title and axes font size and also added the axis labels and units.***

- Fig2: overall font size is very small in these figures, consider increasing these. The choice of colours is also not optimal, especially to people coping with mild colour blindness.

*Response: Thank you for pointing this out. We have adjusted the font sizes and line colours. In aiming to make the revised manuscript more concise, this figure will be transferred to an Appendix, along with some other technical material on the zonal momentum balance.*

Could you explain why the wind stress patterns deviate from those shown in figure 1, is this due to land/sea distribution?

*Response: Yes, Figure 1 shows the applied zonal mean wind stress. Figure 2 shows the integral of the momentum input from wind stress on the ocean, which masks land with zeroes.*

- Fig3: this is a nice overview of the different stream function patterns. As the figure mostly shows the extent/strength of the different gyres, I would redefine the reference value of the stream function.

As shown now, especially the cases with stronger zonal flow are hard to compare as the entire stream function simply becomes more negative. Since you already define the ST/SP gyre boundary, why not use the zonally averaged BSF value as 0 reference?

*Response: The streamfunction is calculated by integrating from a 0 value on the southern boundary. This reference value could be redefined as the reviewer suggests, since it is effectively an arbitrary constant and any value could be selected. This would maintain the facility that contours of streamfunction that are closer together/further apart indicate faster/slower depth-integrated flow. However, it would break many other useful features of the streamfunction. For example, with a zero reference, the transport of a gyre can be identified by picking out the maximum value, whilst the transport through TG and DP can be found by looking for the value on the northern boundary of the choke point. To do this with a non-zero reference value would require knowing what that value is for each panel and correctly taking the difference. Whilst this is fairly simple in itself, it does prevent a quicker and simpler quantification.*

*We have considered using a non-linear colour scale. This would allow for more colours to be used at extreme values. However, it also breaks the perception that equal colour intervals correspond to equal streamfunction intervals. As a result, we decided to retain a linear colour scale.*

*In the revised manuscript, we have used a different colour contours, and typically use white contours and contour labels at extreme negative values of streamfunction may to esolve the issue the reviewer raises (see Page 7, Figure 2).*

- Fig4: The quality of this figure needs to be improved, especially the arrows are unclear and missing a reference . Like figure 4, the entire figure seems to be squeezed vertically, so I would suggest to change the original aspect ratio.

*Response: We have removed the arrows and added contours to improve the clarity of this figure.*

- Fig5: It is interesting to see that the bottom flow component is small, especially with the deeper gateway configurations.

To interpret this, it would be very helpful to show the zonally averaged depth profile of these gateways as used in the model.

***Response: In general, if there is an obstacle large enough to block contours of f/h (Coriolis frequency/depth) by directing them into a continent, then we would expect low bottom velocities as a result of the zero flow condition at the continent. In this case, even deep gateways still fulfill this condition, given the latitude and overall depth of the ocean. We have not introduced this point into the current version of the paper, since it would rely on further dynamical development and increase the length of the paper. The depths used to describe the gateways, e.g., 300m, etc, are already a good indication of their overall depth. We are wary of adding additional figures that would unduly increase the length of the revised paper. Reviewers think the manuscript is too long in length, so we prefer to refer reviewers to Figure D1 in the Appendix (Page 23), which can show depth of gateways.***

- Fig7: this figure is very hard to read due to font and panel sizes, consider e.g. putting 3 cases in a single panel instead.

  ***Response: We have adjusted the structure of this figure as recommended.***

- Fig8: same remarks as for Figure 2.

  ***Response: We have done similar revisions as for Figure 2.***

**Technical comments/typo's**:
- L105: pre-Eocene; do the authors mean the Paleocene or is this a typo?

  ***Response: We have modified this in the revised manuscript; "Page 2, Line 32: …keep Antarctica warm prior to E-O transition."***

- L357: Is the atmospheric pressure (gradient?) zero, or is it negligible compared to the pressure gradients in the ocean?

  ***Response: We do not apply an atmospheric pressure at the ocean surface as part of our forcing. We have revised this sentence to make this clear; "Page 19, Line 27: The transfer of zonal momentum from the atmosphere to the fluid can be neglected as this study does not apply an atmospheric pressure at the ocean surface as part of external forcing."***

- L405: 'Here our'

  ***Response: We have deleted "Here".***

- L421: southward?

  ***Response: Yes.***

- L766: double bracket not needed?

  ***Response: Yes, we have deleted the exterior bracket.***

- L778: 'due to as'

  ***Response: We have deleted "as".***

**Response to referee 2**

The manuscript addresses the question of the Eocene-Oligocene Southern Ocean and its sensitivity to wind stress strengthening and widening/deepening of the Tasmanian Gateway and Drake Passage in setting up an ACC. The paper is very interesting and provides a solid demonstration of the momentum balance at play through an analysis of the zonal momentum balance and its different terms. The results, subject to all possible limitations and caveats, are convincing. However, I found the paper poorly written, very long and repetitive at times. I have the impression the same message and results can be conveyed with perhaps half of the text, improved figures and a more structured discussion/summary.

***We are grateful to the referee for their positive and helpful comments. We have addressed the proposed problems and strengthened our paper by improving its clarity and conciseness. Our response is given below in bold and italic.***

Please find below a list of suggestions, questions and corrections.

L56 New and improved estimates could be used here: Koenig et al. (2014) estimated a full depth transport of 141 ± 2.7 Sv and Chidichimo et al. (2014) and Donohue et al. (2016) estimated a full depth transport of 173.3 ± 10.7 Sv.

***Response: Thanks for suggesting these recent estimations. We have modified this sentence in the revised manuscript; "Page 1, Line 40: The ACC has a volume transport of 127.7 ± 1 Sv (1 Sv=106 m3s−1, Chidichimo et al., 2014) to 137 ± 7 Sv at Drake Passage (DP) (Meredith et al., 2011), 141 ± 13 Sv from in situ and satellite observations (Koenig et al., 2014) or 173.3 ± 10.7 Sv when the near bottom flow is included (Donohue et al., 2016)."***

L115 This is something that you could easily check and should be shown to test the regime change from subtropical gyre dominated to a proto-ACC: please add an analysis of the ocean heat trasport and its eddy contribution.

***Response: We have added our analysis of meridional heat transport in the revised manuscript to show the sensitivity of heat transport to the inception of proto-ACC (see Figure 4; Page 6, Line 23-47). This increases the length of our paper, although we have aimed to keep this new material as concise as possible. However, we feel this extra length is warranted given the increase in clarity and that both reviewers had questions regarding the heat transport. Due to space constraints, we have restricted ourselves to only discussing the total meridional heat transport, which offers a simple explanation for the temperature changes between our experiments.***

L154-155 Please rephrase, something is odd here.

***Response: We have deleted "remains unknown".***

L240 Why do you use a model with no sea-ice? In understand and appreciate the idealized framework of a regional configuration but I don't see what is gained here by eliminating all possible feedbacks induced by sea-ice. Also, the model of Hutchinson et al, 2018 presumably uses sea-ice (CM2.1), so your surface restoring has that infomration.

***Response: Around 38 Ma, the observed global SST was in the range of 25 to 19.5 °C (Bijl et al., 2009; Liu et al.,2009; Houben et al., 2019). The minimum SST of our simulations is around 11°C, which reflects the warm zonal-annual-mean conditions***

*in Hutchinson et al.'s model. These warm surface temperatures remove the possibility of sea-ice in our model, and so we have not included a sea-ice component.*

*(Bijl, P.K., Schouten, S., Sluijs, A., Reichart, G.J., Zachos, J.C. and Brinkhuis, H., 2009. Early Palaeogene temperature evolution of the southwest Pacific Ocean. Nature, 461(7265), pp.776-779.*

*Liu, Z., Pagani, M., Zinniker, D., DeConto, R., Huber, M., Brinkhuis, H., Shah, S.R., Leckie, R.M. and Pearson, A., 2009. Global cooling during the Eocene-Oligocene climate transition. science, 323(5918), pp.1187-1190.*

*Houben, A.J., Bijl, P.K., Sluijs, A., Schouten, S. and Brinkhuis, H., 2019. Late Eocene Southern Ocean cooling and invigoration of circulation preconditioned Antarctica for full-scale glaciation. Geochemistry, Geophysics, Geosystems, 20(5), pp.2214-2234.)*

L248 You use a relatively strong SST and SSS restoring of 10 days. How is that affecting your simulations and results when you try to initiate a thermal isolation of the Antarctic?

*Response: Due to the strong feedback between SST and surface heat flux, 10 days is a fairly standard restoring time for SST in many applications. In contrast, for salinity there is only a weak feedback between SSS and evaporation/precipitation. As a result, this is quite a short timescale for salinity restoring. Our aim in using 10 days for both temperature and salinity was to ensure a good fit to Hutchinson et al.'s SST and SSS fields.*

*In general, a short restoring time scale in the model tends to lead to higher fluxes and acts to damp surface variance in temperature/salinity, as well as eddy kinetic energy. As shown by Zhai & Munday (2014), this also leads to a larger sensitivity to wind stress of the overturning, which is one reason why we have not examined the MOC in this paper. Zhai & Munday do not comment on the sensitivity of the circumpolar transport. However, examination of their figures indicates surface restoring makes the isopycnal slopes less sensitive to wind stress, which implies that the thermal wind transport is as well. In our case, the thermal wind transport is the majority of the zonal flow through TG and DP. Therefore, we can expect that the sensitivity we report is lower than might be achieved in a model that used a pure flux condition on temperature and salinity.*

*In our revised manuscript, we have added a short discussion of the surface restoring, as a response to both reviewers (Page 15, Line 41-69). This includes a short version of the above discussion.*

*(Zhai, X. and D. R. Munday, (2014). Sensitivity of Southern Ocean overturning to wind stress changes: Role of surface restoring time scales, Ocean Modelling, 84, 12-25).*

L281 I am not sure about a regional configuration, but a spin-up of 80 years and sensitivity experiments of 60 years seem a little short to me. It would be intersting to see time series of different metrics to show the circulation is stable and how it changes with the deepening of gateways and shifting of winds.

*Response: The model was spun-up for 85 years by Sauermilch et al. (2021). After changing to our revised wind stress, we ran our experiments for an additional 60 years to adjust to the wind stress conditions. In terms of circumpolar transport, our simulations are well equilibrated. We have added the following figure of a time series*

*of zonal transports to the Appendix of the revised manuscript (see Figure E1). This shows a good degree of equilibration, according to this metric, for most of our experiments.*

[Figure]

L284 Another point related to the model configuration: I am not sure what the actual shape of the zonal wind stress is. Is it a zonal mean and you simply shift it north and south? It is not clear form the text whether zonal wind stress is zonally dependent. Presumably that would matter in terms of alignments with the gateways and relative strength at the DP and TG.

*Response: We apologise for the lack of clarity on the form of the wind stress. It is, indeed, a zonally-symmetric wind stress. It has been smoothed slightly, relative to Sauermilch et al. (2021), in order to make adjusting it to the north and south cleaner. We will make this point clearer during revision (see Page 5, Line 13).*

*A non-zonally-symmetric wind stress would indeed raise interesting questions regarding the alignment with TG and DP. We have raised this point in the discussion of our revised manuscript (Page 16, Line 44), thank you for suggesting this idea.*

L360 details of the discretization, also in L634, should go into the supplmentary information (Eq. 4 is already present). Also, Eq. S3 is missing the 1/\rho_0.

*Response: Thanks for spotting the missing 1/rho_0. We have added more detail of the algorithm to the Appendices of the revised paper. The revised paper reads, at this point; "Page 12, Line 1: Following Masich et al. (2015), the zonally vertically integrated total zonal pressure gradient (or total topographic form stress) is discretised as per Eq (4). We extract the total topographic form stress from the zonally vertically integrated total zonal pressure gradient field. More detail on the calculation of topographic form stress, and errors associated with the use of partial model cells, can be found in Appendices B and C"*

L415 Fig.4 is really difficult to read with its present choice of coulours and arrows and should be improved. Consider a specific countour for the SST to highlight the change in temperature along the coast, and different/fewer arrows. Also perhaps less panels

*Response: We have removed the arrows and added the suggested contour to improve the clarity of this figure (see Page 8, Figure 3)*

L478 Eg. 5 is missing

*Response: Thanks for spotting the missing equation, we have adjusted the numbers appropriately.*

Figure 7 This figure is also difficult to read. Consider adding to the same panel both the normal and doubled wind stress to highlight differences.

*Response: Thank you for the idea on how to modify this figure. We have combined the panels as suggested and altered the number of lines so as to improve clarity (see Page 13, Figure 7).*

L632 Eq. 5 is missing as well as section 2.5

*Response: Thank you for spotting these errors. We have adjusted the numbers appropriately.*

L665 I really like your results but the Discussion section is difficult to read, repetitive and often not a 'Discussion' but rather a 'Summary'. Please improve your text to ease the read.

*Response: Thank you for your positive opinion. We have improved the structure and flow of the Discussion. We have removed any repeated material and seek to improve its clarity (see Page 15 - 17).*